# Longitudinal, Multi-Platform Metagenomics Yields a High-Quality Genomic Catalog and Guides an *In Vitro* Model for Cheese Communities

Christina C. Saak,[a] Emily C. Pierce,[a] Cong B. Dinh,[a] Daniel Portik,[b] Richard Hall,[b] Meredith Ashby,[b] Rachel J. Dutton[a]

[a]Division of Biological Sciences, Section of Molecular Biology, University of California San Diego, La Jolla, California, USA
[b]Pacific Biosciences, Menlo Park, California, USA

**ABSTRACT** Microbiomes are intricately intertwined with human health, geochemical cycles, and food production. While many microbiomes of interest are highly complex and experimentally intractable, cheese rind microbiomes have proven to be powerful model systems for the study of microbial interactions. To provide a more comprehensive view of the genomic potential and temporal dynamics of cheese rind communities, we combined longitudinal, multi-platform metagenomics of three ripening washed-rind cheeses with whole-genome sequencing of community isolates. Sequencing-based approaches revealed a highly reproducible microbial succession in each cheese and the coexistence of closely related *Psychrobacter* species and enabled the prediction of plasmid and phage diversity and their host associations. In combination with culture-based approaches, we established a genomic catalog and a paired 16-member *in vitro* washed-rind cheese system. The combination of multi-platform metagenomic time-series data and an *in vitro* model provides a rich resource for further investigation of cheese rind microbiomes both computationally and experimentally.

**IMPORTANCE** Metagenome sequencing can provide great insights into microbiome composition and function and help researchers develop testable hypotheses. Model microbiomes, such as those composed of cheese rind bacteria and fungi, allow the testing of these hypotheses in a controlled manner. Here, we first generated an extensive longitudinal metagenomic data set. This data set reveals successional dynamics, yields a phyla-spanning bacterial genomic catalog, associates mobile genetic elements with their hosts, and provides insights into functional enrichment of *Psychrobacter* in the cheese environment. Next, we show that members of the washed-rind cheese microbiome lend themselves to *in vitro* community reconstruction. This paired metagenomic data and *in vitro* system can thus be used as a platform for generating and testing hypotheses related to the dynamics within, and the functions associated with, cheese rind microbiomes.

**KEYWORDS** microbiomes, multi-platform metagenomics, mobile genetic elements, pangenomics, amplicon sequencing, Oxford Nanopore sequencing, PacBio HiFi sequencing, metaHi-C, washed-rind cheeses, *in vitro* microbiomes

Address correspondence to Rachel J. Dutton, rjdutton@ucsd.edu.

The authors declare a conflict of interest. Authors Daniel Portik, Richard Hall and Meredith Ashby were employees of Pacific Biosciences at the time of the study. Pacific Biosciences sponsored the HiFi sequencing by providing reagents and technical support.

Microbiomes play crucial roles in human health (1), geochemical cycles (2), and food production (3). While the characterization of community composition of diverse microbiomes has come a long way, our mechanistic understanding of community functioning, and thus our ability to predict and manipulate it, lags behind (4). Given the complexity and experimental intractability of many microbiomes of interest, model microbiomes consisting of a manageable number of community members which are experimentally tractable can help facilitate the generation and testing of hypotheses about microbiome function. Cheese rind communities have already provided valuable

insights into the biology of microbiomes and the microbial interactions within them as they provide tractable systems for both *in situ* and *in vitro* studies (5). Building on this insight, communities have been assembled to show that genetic requirements change in microbial communities of increasing complexity, partially driven by the cross-feeding of amino acids and the competition for iron (6). In fact, competition for iron has been repeatedly shown to impact microbial interactions within cheese rind microbiomes (7, 8). Other important aspects of microbial ecology have also been investigated using model cheese rind microbiomes, including the impact of strain-level diversity on community assembly and function (9), the importance of physical networks provided by filamentous fungi in the community for the dispersal of bacterial community members (10), and the effect of fungal volatile compounds on microbiome assembly (11). The cheese rind model communities currently available for *in vitro* studies are based on natural (5) and bloomy rind communities (6) and represent low-complexity microbiomes ranging from 3 to 7 species (5, 6, 12, 13).

In contrast to natural and bloomy rind cheeses, washed-rind cheeses are produced by regular washing (or smearing) with a brine solution (14). As such, the microbial communities on the surface of a washed-rind cheese experience homogenization throughout its development, which may facilitate intermicrobial interactions or evolutionary processes such as horizontal gene transfer (HGT). To date, several studies have examined the community composition of washed-rind cheeses using culture-dependent and culture-independent techniques. For example, several studies have isolated bacterial community members and subjected the isolates to 16S amplicon sequencing for taxonomic classification and/or whole-genome sequencing (15, 16). Another study employed 16S amplicon sequencing to taxonomically characterize the bacterial members of a Belgian washed-rind cheese microbiome (17). A number of other studies have additionally used ITS amplicon sequencing to taxonomically characterize the fungal community portion (18–20). To investigate how the microbial communities change over time, several studies have combined this 16S/ITS amplicon sequencing with longitudinal sampling (21, 22). In the past, shotgun metagenomic sequencing has also been used to gain functional insights into these communities (23). Together, these studies have shown that bacteria often outnumber fungi by orders of magnitude (21, 22). Among the bacterial community members, Actinobacteria, such as *Brevibacterium*, *Corynebacterium*, and *Glutamicibacter*, and Proteobacteria, such as *Psychrobacter* and *Halomonas*, are frequently detected together in these communities. For example, several studies detected all five of these genera (15, 17), while in another study, all except for *Corynebacterium* were detected above the threshold (21). Yet another study found all of the genera mentioned above except for *Glutamicibacter* (22). Lastly, it has been shown that the communities associated with these cheeses show reproducible community succession (21). Emerging metagenomic techniques have the potential to enable an even deeper characterization of the species- and strain-level diversity, functional potential, eco-evolutionary dynamics, and mobile genetic elements (MGEs) within these communities.

In this study, we combine several metagenomics techniques (amplicon, short-read and long-read shotgun sequencing, and metaHi-C) with a longitudinal data set of three washed-rind cheese communities that were collected over the course of cheese ripening. These data provided insights into the reproducible successional trajectories of the studied washed-rind cheeses, provided a catalog of genomes of community members that can be used as references for future studies, identified plasmids and phages and associated them with their hosts, and provided insights into the biology of these communities. For example, we investigated the striking diversity of *Psychrobacter* genomes recovered from the communities and present a functional enrichment study that highlights the enrichment of genes involved in type VI secretion and siderophore acquisition, two traits which are of high value in the densely populated, iron-limited environment of the cheese rind. Finally, we used the genomic catalog to establish a representative culture collection of washed-rind community members and reconstituted an *in vitro* model community based on the washed-rind cheese microbiome. This model microbiome contains 16 members from

several microbial phyla, including several *Psychrobacter* representatives, allowing the examination of interactions across diverse microbial groups and representing the most complex cheese rind-based model microbiome to date. We show that this *in vitro* community undergoes reproducible succession and, by removing certain taxonomic groups, we begin to investigate the interaction dynamics in this community.

## RESULTS

**Longitudinal sampling of three washed-rind cheeses from the same facility.** Rind samples of three different types of washed-rind cheeses (Cheese A, B, and C) were collected from triplicate batches (each batch made approximately 1 week apart) at six time points throughout ripening (Fig. 1A, Fig. S1A, see Table S1). Cheeses A, B, and C differ with respect to their milk source, use of pasteurized versus raw milk (pasteurized milk for Cheeses A and B, raw milk for Cheese C), production location, and production methods. All three cheeses were aged in the same facility and subjected to similar aging practices, such as repeated washing with a brine solution, albeit at different intervals (Table S1). For all samples, DNA was extracted and 16S and ITS amplicon sequencing were performed. Subsequently, samples from Batch 3 were characterized using in-depth metagenomic sequencing, including short-read metagenomic sequencing of all six time points; long-read metagenomic sequencing of rinds at weeks 2, 3, 4, 9 and 13; and metaHi-C at weeks 2, 4, and 13 in the case of Cheeses B and C and at weeks 2 and 13 in the case of Cheese A (Fig. S1B, Tables S1 and S2).

**Successional dynamics of the rind communities throughout ripening.** To gain a higher-level overview of community dynamics and reproducibility of succession patterns, we analyzed relative abundances within bacterial and fungal populations using 16S and ITS sequencing (Tables S3 and S4). The successional dynamics of each of the three cheeses were remarkably reproducible for both the bacterial and the fungal communities (Fig. 1B to D). Principal-component analysis based on Bray-Curtis indices indicates that while the different cheeses were highly similar at the earliest sampled time points, they diverged from each other along reproducible trajectories throughout the aging process (Fig. 1B). One exception was one batch of Cheese B, which clustered more closely to Cheese A at the final fungal sequencing time point. This difference is mainly due to *Fusarium*, which was found in all 3 batches of Cheese A and detected in the ITS sequences of that batch of Cheese B, but not found in the other two batches of that same cheese (Fig. 1D). Indeed, one wheel from this batch had a visibly different rind than the cheese wheels from the other two batches (Fig. S2A). Although the three communities diverged over time at the genus level (Fig. 1B), there are consistent successional patterns at the phylum and order levels across all cheeses (Fig. S2B to C). All three cheeses were dominated by Firmicutes in week 1, likely due to the lack of rind resulting in sampling of lactic acid bacteria in the cheese core. Proteobacteria quickly took over and dominated the bacterial communities of all three cheeses by the end of ripening. Cheese A also shows a reproducible establishment of Actinobacteria and Cheese C additionally shows a reproducible establishment of Bacteroidetes in the community (Fig. 1C, Fig. S2B). Regarding the fungal community members, all three cheeses were dominated by Saccharomycetales throughout the ripening process, while Hypocreales reproducibly established in the communities of Cheeses A and C, but not Cheese B, over time (Fig. 1D, Fig. S2C).

While amplicon sequencing provides a high-level overview of community succession and its reproducibility, it does not address the relative abundance of bacteria and fungi to each other. To close this information gap, we performed taxonomic classification of long-read shotgun metagenomic sequencing data (Table S5). The majority of reads were classified to at least the genus level (Fig. S3A). Long-read-based taxonomic classifications revealed that all three cheeses were heavily dominated by bacteria and that fungi only constituted a small proportion of the communities, especially at the end of ripening (Fig. 2, Fig. S3B). Consistent with the amplicon sequencing, long-read based taxonomy revealed genera which were shared between all three cheeses, such as *Psychrobacter* and *Pseudoalteromonas*; as well as genera which were specific to

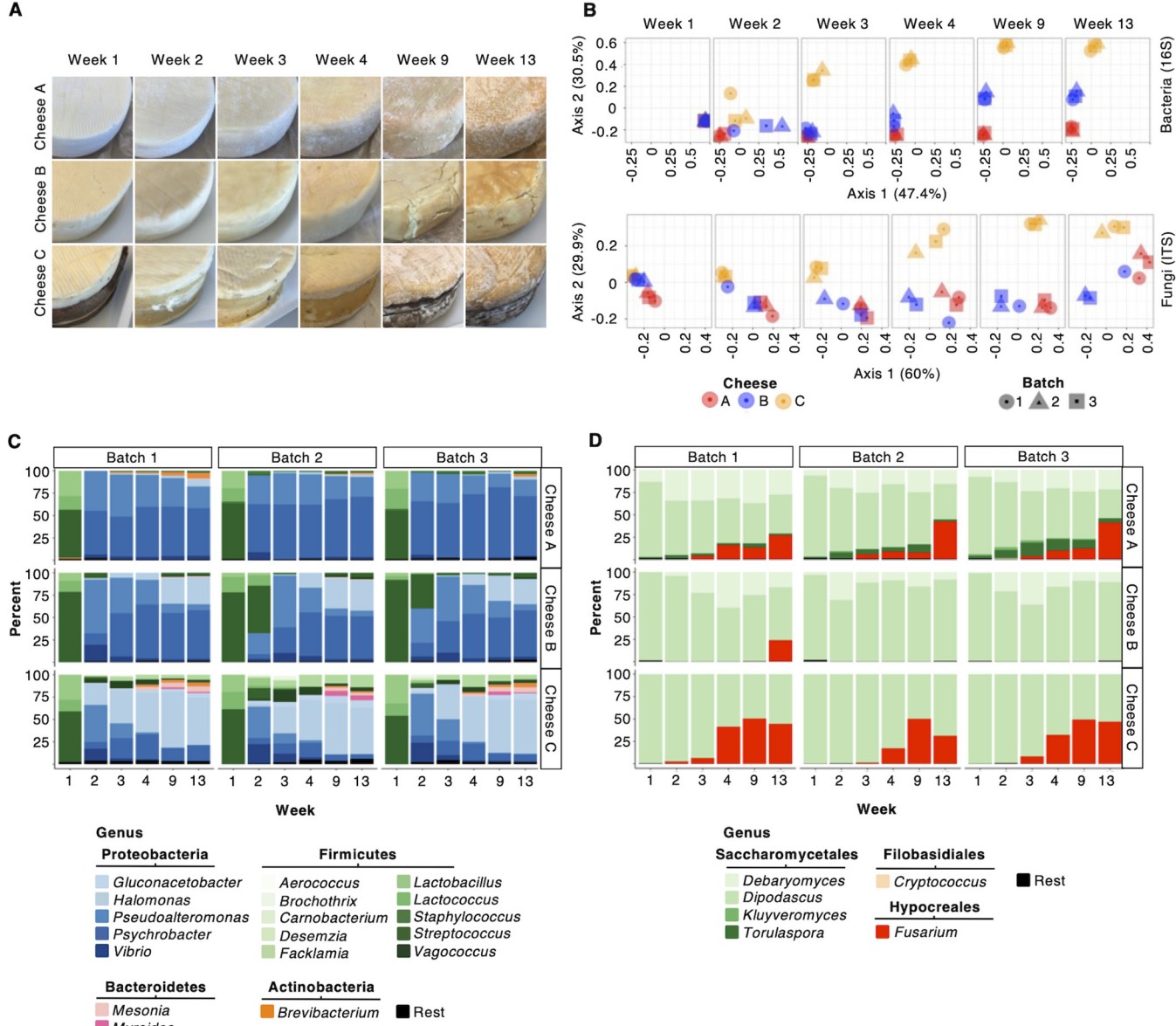

**FIG 1** Three washed-rind cheeses from the same facility show reproducible succession patterns during ripening. (A) For each of the three different washed-rind cheeses (A, B, and C), we followed the aging of three different batches produced 1 week apart. From each batch, we collected rind from duplicate wheels at six time points. Detailed sequencing overview and collection schedules are found in Fig. S1 and Table S1, respectively. Representative images of each of the three cheeses at different time points are shown. Cheese C is wrapped in spruce during ripening. In the pictures for weeks 2 to 4, the spruce has been removed. (B) 16S and ITS data of the ripening communities were subjected to principal-component analyses of Bray-Curtis dissimilarities. Colors correspond to different cheeses; shapes correspond to different batches. (C and D) Relative abundance plots of (C) bacteria as determined by 16S amplicon sequencing and (D) fungi as determined by ITS sequencing. Relative abundances of amplicon sequence variants collapsed at the genus level are shown. Rest = genera with <1% of the classified reads. Detailed information about 16S and ITS read statistics can be found in Tables S3 and S4, respectively.

certain cheeses, such as *Alcaligenes* and *Sphingobacterium*, which were specific to Cheese C (Fig. 2, Fig. S3B).

Overall, Cheese C contained the largest number of unique taxa per rank at all sampled time points (Fig. S3C), which is in accordance with the amplicon results. Cheese C also showed the highest degree of taxa turnover (Fig. 2, Fig. S3B); it was initially dominated by *Pseudoalteromonas*, which was then overtaken by *Halomonas* and *Alcaligenes*. In addition, Gram-positive community members such as *Brevibacterium* and *Sphingobacterium* become more abundant over time in Cheese C than they did in the other two communities. In contrast, Cheese B was initially dominated by the yeast *Debaryomyces* before a bacterial community dominated by *Psychrobacter* and *Pseudoalteromonas* took over. Eventually,

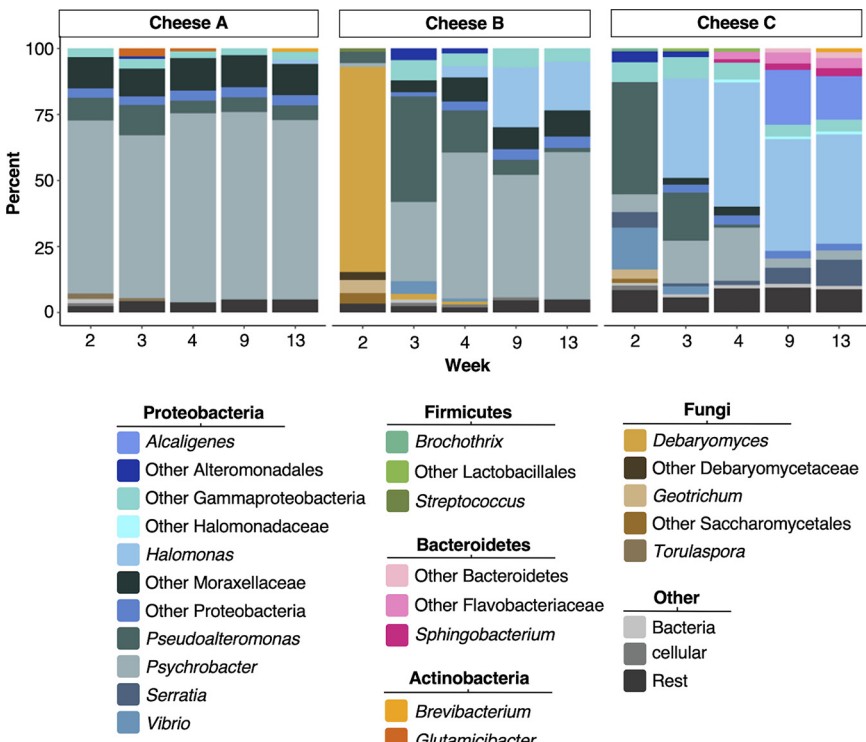

**FIG 2** Long-read-based relative abundance reveals dominance of bacterial community members. Relative abundance plots of taxa (aggregated at the genus level or higher) detected by long-read shotgun metagenomic sequencing. Rest = taxa with <1% of the classified reads. Detailed information about read statistics can be found in Table S5.

*Pseudoalteromonas* was largely displaced from the Cheese B community, while *Halomonas*, members of the family Moraxellaceae, and other unidentified Proteobacteria became more abundant. In contrast to both Cheeses B and C, Cheese A showed very minimal community succession between weeks 2 and 13 and was dominated by *Psychrobacter* and other members of Moraxellaceae throughout this entire period of ripening.

**A phyla-spanning genomic catalog of washed-rind cheese communities.** We next aimed to generate a genomic catalog for each of the three cheeses to help provide insights into the sub-genus diversity, functional potential, and MGE diversity of the communities. To do so, we assembled the long-read shotgun data for each cheese, either by time point or across all time points (co-assemblies) (Table S6). The assemblies were then binned to generate metagenome-assembled genomes (MAGs) (Table S7). In addition, community members from Cheese B were isolated and subjected to short- and long-read sequencing for *de novo* hybrid genome assembly (Table S8). For each cheese, we combined all high-quality MAGs from the individual time point assemblies, co-assemblies, and isolate genomes (for Cheese B). We then dereplicated each data set and selected representative MAGs for each cheese (Table S9). For selection of representative MAGs/genomes from a given dereplication group, isolate genomes were given the highest priority, followed by circular MAGs from individual time point assemblies, then circular MAGs from co-assemblies, then complete, noncircular MAGs from individual time point assemblies, and finally complete, noncircular MAGs from co-assemblies. If several bins per group fell into the same category, the MAGs were prioritized based on quality as assessed by CheckM.

For Cheeses A and C, we recovered 17 and 37 high-quality MAGs, respectively (Fig. 3, Table S9). In Cheese A, 12 of the 17 high-quality MAGs were both single-contig and circular, while the other 5 noncircular bins contained 2 to 6 contigs each. In the case of Cheese C, 24 of the 37 high-quality MAGs were single-contig and 19 of those

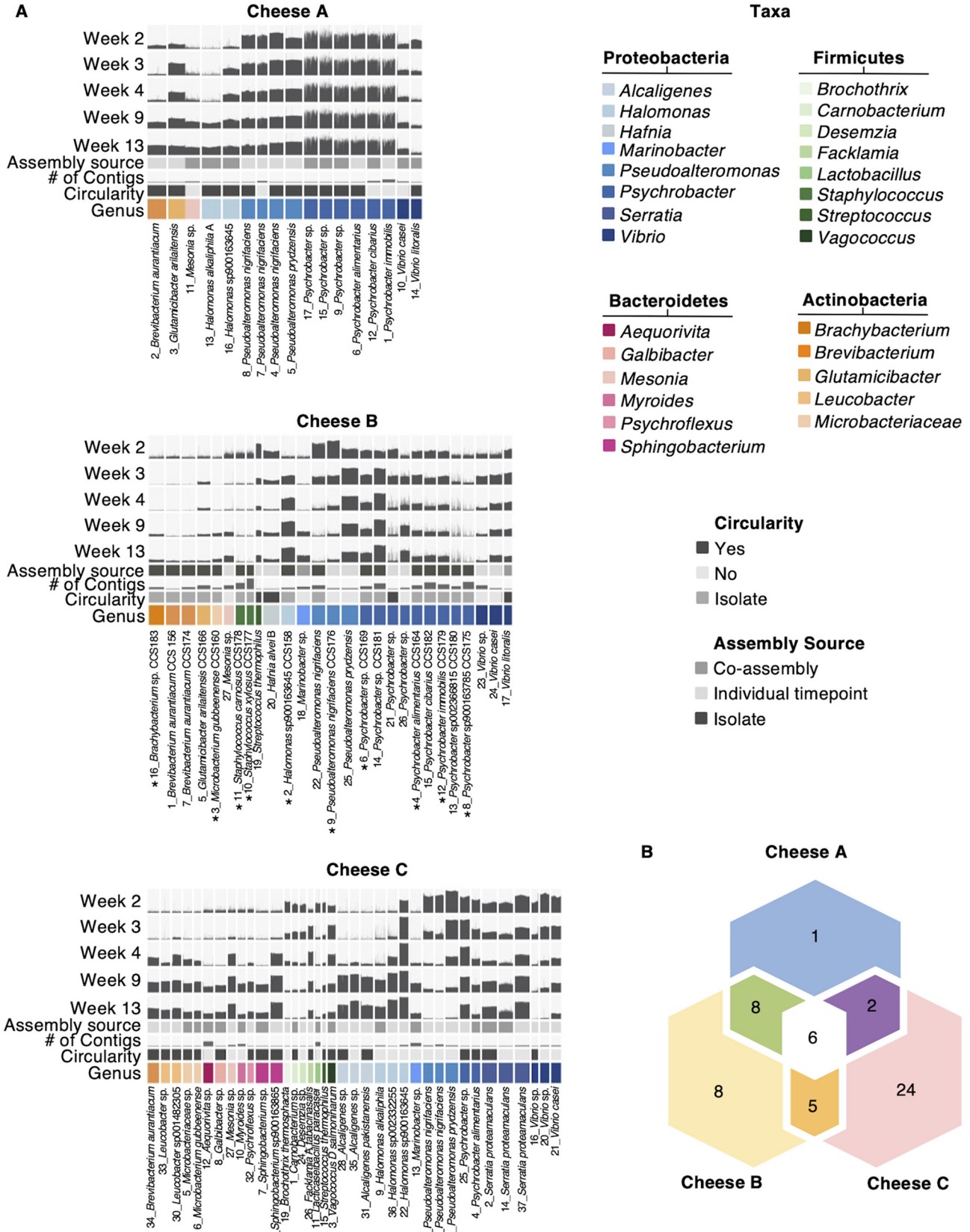

**FIG 3** A phyla-spanning bacterial genomic catalog of three washed-rind cheeses. (A) Anvi'o plots show dereplicated high-quality metagenome-assembled genomes (MAGs) (and isolate genomes in the case of Cheese B). MAGs were generated either by assembling HiFi reads from individual

were circular. The rest of the bins from Cheese C contained 2 to 10 contigs each. For Cheese B, we recovered 11 high-quality MAGs and 16 isolate genomes (Table S9). Of these 11 high-quality MAGs, 4 were both single-contig and circular. The other 7 noncircular bins contained 2 to 8 contigs each.

Because both the amplicon sequencing data and long-read data indicated an overlap between the genera present in the three communities, we next tested whether identical or highly similar genomes were recovered from these communities as well. Indeed, comparing the genomic catalogs recovered from the three cheeses using average nucleotide identity (ANI) values revealed that the cheeses contained both common and unique genomes (Fig. S4, Table S10). Specifically, when considering an ANI cutoff of 99, we observed that 6 MAGs were represented in the genomic catalogs from all three cheeses (Fig. 3B). Based on the Genome Taxonomy Database Toolkit (GTDB-Tk) (24), these MAGs were annotated as *Mesonia* spp., *Vibrio casei*, *Pseudoalteromonas nigrifaciens*, *Psychrobacter alimentarius*, *Vibrio litoralis*, and *Pseudoalteromonas pyrdzensis*. Additionally, Cheeses A and B had 8 MAGs (and isolate genomes) in common, Cheeses A and C shared 2 MAGs, and Cheeses B and C shared 5 MAGs (and isolate genomes) (Fig. 3B, Fig. S4, Table S10). Lastly, Cheese A had 1 unique MAG, Cheese B had 8 unique MAGs (and isolate genomes) and Cheese C had 24 unique MAGs (Fig. 3B, Fig. S4, Table S10).

Altogether, the genomic catalog covers the bacterial phyla Proteobacteria, Firmicutes, Bacteroidetes, and Actinobacteria (Fig. 3A, Table S9). For each cheese, Proteobacteria was the group with the most MAGs (and isolate genomes), with 14, 18, and 18 MAGs (and isolate genomes) representing this phylum in Cheeses A, B, and C, respectively. Furthermore, we were able to recover multiple species and/or strain representatives for several genera. For Cheeses A and B, we observed multiple co-occurring *Psychrobacter* species and/or strains. For Cheeses A and B, 6 out of 14 and 9 out of 18 Proteobacteria, respectively, were from the genus *Psychrobacter*. Both cheeses also contained a diversity of distinct *Pseudoalteromonas* representatives, with 4 and 3 distinct MAGs (and isolate genomes) recovered from Cheeses A and B, respectively. Cheese C also contained 2 and 3 distinct MAGs belonging to the genera *Psychrobacter* and *Pseudoalteromonas*, respectively. In addition, we observed species and/or strain-level diversity in the genera *Alcaligenes*, *Brevibacterium*, *Halomonas*, *Leucobacter*, *Microbacterium*, *Serratia*, *Sphingobacterium*, *Staphylococcus*, and *Vibrio*. We further noted that 4 of 17, 8 of 27, and 15 of 37 MAGs (and isolate genomes) from Cheeses A, B, and C, respectively, were not classified at the species level by GTDB-Tk (Table S9), which could indicate potentially new species in our genomic catalog.

To gain an overview of how abundant each of the recovered MAGs (and isolate genomes) was during community development, the communities were also subjected to short-read metagenomic sequencing (see Table S2), and the reads were mapped to the genomes in the genomic catalog (Fig. 3A). For almost all time points, more than 50% of the reads were mapped to the genomic catalog (see Table S11). The only exception was Cheese B on week 2, in which only about 19% of the reads were mapped to the genomic catalog. From the long-read based community composition analysis (Fig. 2), we know that Cheese B was dominated by a fungus at this time point. Because the genomic catalog shown in Fig. 3A only contains bacterial genomes, it is not surprising that a small amount of short reads from week 2 mapped against the bacterial genomic catalog for Cheese B. In contrast, for some time points, over 90% of the short reads from Cheeses B and C mapped to the genomic catalog (Table S11), indicating that the catalog provides a good cross-section of the bacterial communities at these time points.

Next, the availability of strain-resolved genomic catalogs allowed us to interrogate

**FIG 3** Legend (Continued)

time points or by co-assembling reads from all time points of a cheese. For each MAG, the number of contigs is indicated (lowest value = 1, highest value = 22). Circularity of MAGs is indicated as is the taxonomy as predicted by GTDB-Tk. Colors correspond to genera. Bin names contain the genomic catalog bin number and the predicted taxonomy of the bin. To estimate relative abundances of MAGs over time, the short-reads from weeks 2, 3, 4, 9, and 13 were mapped to the genomic catalog for each cheese. Bar height indicates the number of reads mapped to the respective genomic region. Table S7 in the supplemental material shows all non-dereplicated bins. Genomes marked with an asterisk (*) were included in *in vitro* community reconstruction. (B) Overlap of high-quality MAGs (and isolate genomes) from Cheeses A, B, and C, with MAGs having an average nucleotide identity (ANI) of >99% considered to be the same.

the successional dynamics of each cheese more deeply. For example, amplicon and long-read-based taxonomy suggested that there were relatively small changes in community composition over time in Cheese A. However, short-read mapping back to the genomic catalog revealed species- and strain-level temporal dynamics (Fig. 3A). Specifically, *P. nigrifaciens* initially slightly dominated over *P. prydzensis* in Cheese A, while by the end of ripening at week 13, *P. prydzensis* slightly dominated over *P. nigrifaciens*. Similarly, *V. casei* initially dominated over *V. litoralis* in Cheese A; while already by week 3, *V. litoralis* began to dominate over *V. casei*. Very similar dynamics were also observed in Cheese B for *Pseudoalteromonas* and *Vibrio* (Fig. 3A). *P. nigrifaciens* initially dominated over *P. prydzensis* before finally being overtaken by it. Similarly, *V. litoralis* initially dominated over *V. casei* before *V. casei* caught up in terms of relative abundance toward the end of ripening. For both Cheeses A and B, we saw little variation in abundances across the *Psychrobacter* MAGs (and isolate genomes), potentially due to a high similarity between the MAGs (and isolate genomes) or the stable coexistence of species/strains over time. For Cheese C, the abundance of *Pseudoalteromonas* MAGs decreased over time; however, their relative abundance to each other did not change significantly (Fig. 3A). For *Psychrobacter*, only one of the two representative MAGs decreased in abundance over time, while the other showed a stable abundance based on read mapping.

**Meta-HiC and long reads associate viruses and plasmids with their hosts.** In addition to generating a genomic catalog for each cheese, we were interested in leveraging existing long-read metagenomics data to characterize the diversity of plasmids and viruses within these microbiomes. To help capture any short and/or low-abundance MGEs which might have been missed in the long-read assemblies, we first generated mega-assemblies of each cheese combining both long-read and high-depth short-read data sets (see Tables S2 and S6). In brief, the short metagenomic shotgun reads were mapped to the co- and individual time point assemblies, and any reads which did not map were assembled. The resulting short read-based contigs were combined with the dereplicated unbinned long read-based contigs and the contigs from the dereplicated MAGs to yield mega-assemblies for each of the three cheeses. These mega-assemblies were subjected to plasmid and virus prediction (Fig. S5, Table S12). We identified 419, 297, and 343 putative plasmid contigs in Cheeses A, B, and C, respectively. Additionally, we identified 109, 115, and 212 lysogenic and 36, 29, and 62 lytic virus contigs for Cheeses A, B, and C, respectively. Of these, 4, 4, and 7 contigs were classified as complete, circular viruses while 22, 32, and 57 contigs were classified as high-quality draft viruses for Cheeses A, B, and C, respectively (Fig. S5). Some contigs were classified as both lytic virus and plasmid. Finally, we considered the size distributions of the predicted viruses (Fig. S6) and plasmids (Fig. S7). In this regard, Cheese C stands out due to the presence of several contigs larger than 200,000 bp, predicted to be viruses (Fig. S6). Sizes above 200,000 bp are a hallmark of jumbophages (25).

We next used the Hi-C data generated for various time points (Table S2) to identify putative hosts for the extrachromosomal MGEs (plasmids and lytic viruses). MGE predictions were used as inputs for the viralAssociationPipeline (26, 27) program together with the total mega-assemblies, metaHi-C reads, and long reads (Fig. 4, Fig. S8, Table S13). In total, 70, 21, and 120 of the predicted extrachromosomal MGEs were associated with MAGs, while an additional 209, 105, and 89 extrachromosomal MGEs were associated with non-MAG (unbinned) contigs for Cheeses A, B, and C, respectively (Fig. S8). For Cheeses A, B, and C, 7, 1, and 11 MGEs, respectively, were associated with more than one MAG (Table S13). The vast majority of putative MGEs were not associated with a putative host through the initial binning (Fig. 4). A total of 173, 199, and 195 of the extrachromosomal contigs remained unassociated.

Of the extrachromosomal MGEs associated with MAGs, we identified 41, 11, and 97 instances of MGEs for Cheeses A, B, and C, respectively, as being associated with a MAG (or isolate genome) at one time point only (Table S13). We further identified 31, 10, and 23 instances of MGEs from Cheeses A, B, and C, respectively, being associated

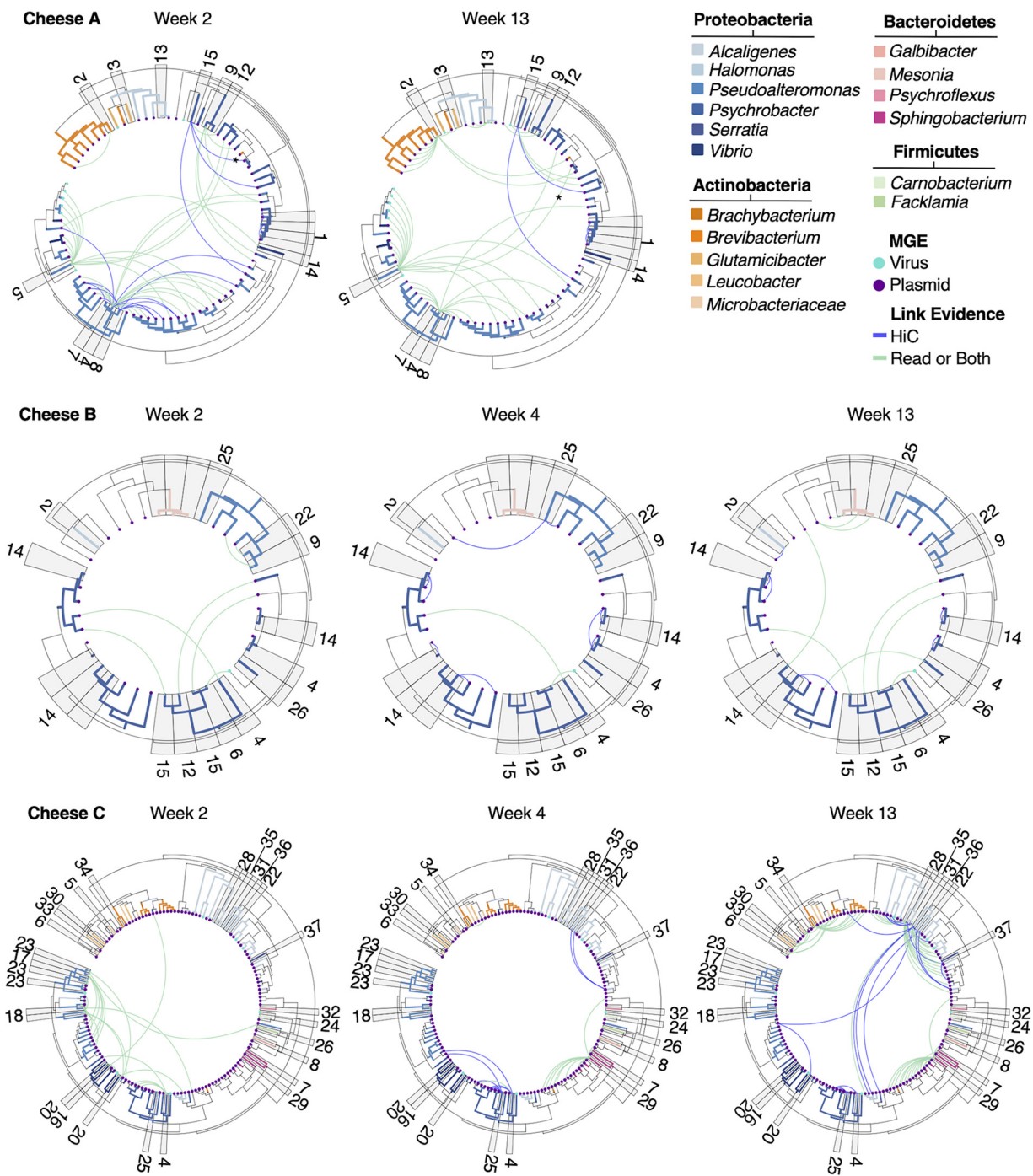

**FIG 4** metaHi-C and long read-based evidence of MGE associations with hosts from the washed-rind cheese genomic catalog. iTOL graphs show the long read-based or metaHiC-based association of MGEs (lytic viruses and plasmids) with MAGs (and isolate genomes) from the genomic catalog. Tree branches represent contigs and are colored by the contigs' predicted taxa. Contigs which are classified as lytic viruses and plasmids are indicated with colored dots (teal = virus, purple = plasmid). Bin numbers for MAG host contigs are shown on the outside of the tree. Associations between MGEs and their hosts are indicated by the lines. Blue lines indicate metaHiC-based evidence for association; green lines indicate long read-based evidence or both metaHiC- and long read-based evidence. An asterisk (*) indicates the associations for contig s673.ctg001008l_6. Full results from the viralAssociationPipeline.py program can be found in Table S13.

with the same MAG (or isolate genome) at two time points, and one instance each of an MGE from Cheese B and Cheese C being associated with the same MAG at three time points. One MGE, a predicted plasmid (s673.ctg001008l_6, indicated by an asterisk in Fig. 4), was associated with two different MAGs from Cheese A at two different time points (Tables S13 and S14), indicating a putative HGT event. At week 2, this MGE is

associated with a *Psychrobacter* sp. host, and at week 13, it is associated with a *P. prydzensis* host. Interestingly, this plasmid is predicted to encode genes involved in iron uptake, which have previously been associated with horizontally transferred regions in cheese (28). The full results of the association pipeline are listed in Table S13.

Next, we more closely examined MGEs which changed their genus association over time (see Table S14). A total of 12, 5, and 4 MGEs from Cheeses A, B, and C, respectively, changed hosts over the course of sampling based on taxonomic prediction of the host contig. We identified several predicted instances of inter-kingdom host changes, especially in Cheese B between *Geotrichum* and proteobacterial species. Again, we identified iron-uptake genes associated with potentially transferred elements. Some of these elements also contained genes related to phosphonate transport.

**Pangenome analysis of *Psychrobacter* and functional enrichment in cheese isolates.** Our genomic catalog revealed high sub-genus-level diversity within the genus *Psychrobacter* (Fig. 3A, Table S9). To investigate conserved and unique gene sets across *Psychrobacter* species, we analyzed the *Psychrobacter* pangenome. We first determined the nonredundant *Psychrobacter* isolate genomes and MAGs from the combined three-cheese data set. This resulted in a total of 7 isolate genomes and 2 MAGs. An additional 97 publicly available *Psychrobacter* genomes were combined with these 9 genomes in a pangenomic analysis (Fig. 5A, see Table S15). The 97 publicly available genomes were sourced from marine, soil, cheese, other fermented food, host-associated, and other miscellaneous environments. We attempted to include at least one representative of every *Psychrobacter* species possessing a publicly available genome. As expected, pangenomic analysis identified a core set of genes common to *Psychrobacter* from diverse environments (Fig. S9). This core gene set, defined as being present in 95% of genomes, consisted of 1,405 genes and made up 9.4% of total gene clusters, with the remaining 90.6% of gene clusters classified as accessory by panX (Fig. S9). A core-genome single-nucleotide polymorphism (SNP) tree constructed from all variable positions of all single-copy core genes showed some evidence of clustering of genomes by environment and, with the exception of *Psychrobacter immobilis*, by species (Fig. 5A). These data suggest that *Psychrobacter* species may have environment-specific gene sets. Functional enrichment analysis of gene clusters was used to find functions (clusters of orthologous groups [COGs]) that were enriched in genomes of cheese isolates compared to genomes from other environments (Fig. 5B). Specifically, a group of genes related to iron access, particularly through the use of iron-chelating siderophores, was enriched in *Psychrobacter* genomes from cheese (adj. q-value < 0.1; Fig. 5B, see Table S16). In addition, genes related to type VI secretion systems, contractile defense systems which bacteria can use to transport effector proteins into target cells, were also enriched in cheese *Psychrobacter* genomes relative to that in other environments (adj. q-value < 0.1; Fig. 5B, Table S16).

Because our metagenomic sequencing indicated that multiple *Psychrobacter* species can coexist within a single cheese (Fig. 3A), we wanted to investigate the genetic basis of this coexistence. As such, we performed an additional pangenome analysis focused specifically on the 8 *Psychrobacter* genomes from Cheese B (MAG and isolate genomes) (Fig. 5C). This analysis identified a set of 'cloud' gene clusters unique to each genome, 'shell' gene clusters found in at least 2 (but not all) genomes, and 'core' gene clusters found in all eight genomes. Because accessory gene sets may provide clues to the coexistence of multiple closely related species or strains, we examined the COG functional categories associated with the core, shell, and cloud gene sets. While central metabolic processes such as amino acid metabolism and translation were most prevalent in the core set, the cloud set was enriched in defense mechanisms and the mobilome (phages and transposons) (Fig. 5D, Table S17). Because these gene categories can be involved in interspecies interactions, this may be an interesting starting point for future experiments to investigate the mechanisms underlying species and/or strain co-occurrence.

***In vitro* community reconstruction.** To facilitate experimental follow-up studies, for example, investigation of *Psychrobacter* interspecies interactions, we next aimed to establish whether the washed-rind cheese communities lent themselves to *in vitro* experimentation. To this end, we selected 16 microbial species (13 isolates from Cheese B, and 3 from

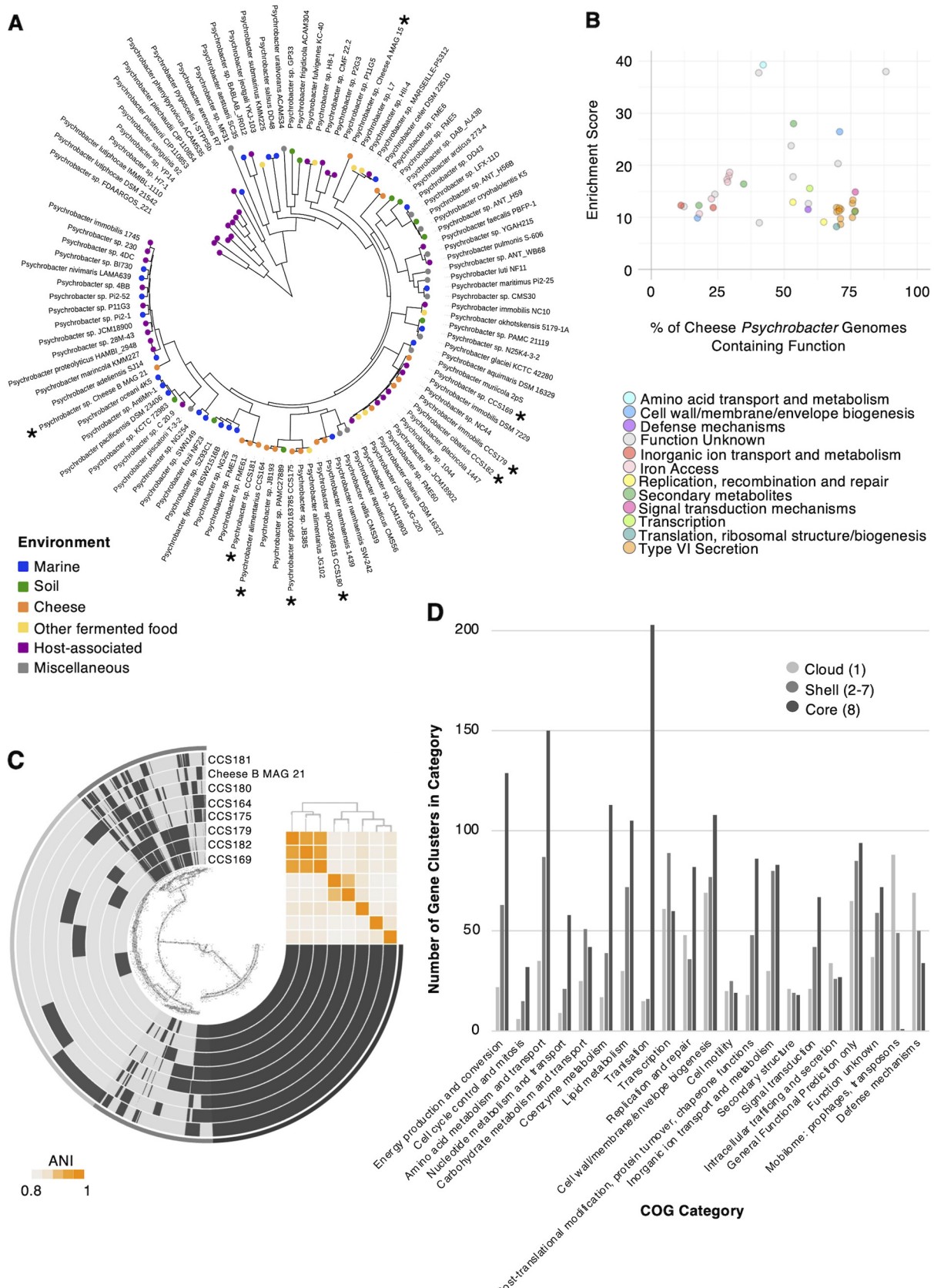

**FIG 5** Pangenomic analysis of *Psychrobacter* genomes from diverse environments. (A) Pangenomic analysis was performed on 106 genomes from marine, soil, cheese, other fermented food, and miscellaneous other environments. The phylogenetic tree was constructed

previous isolation efforts) which would represent both the breadth and depth of diversity found in a typical washed-rind cheese microbiome based on our own and previous analyses. Given that the metagenomic analysis presented here revealed that multiple species and strains of the genus *Psychrobacter* can co-occur, we chose to include 4 *Psychrobacter* isolates from Cheese B: *P. alimentarius* CCS164, *Psychrobacter* sp. CCS169, *Psychrobacter sp900163785* CCS175, and *P. immobilis* CCS179. These isolates represent two different groups of *Psychrobacter* (Fig. 5A and C). In addition to the *Psychrobacter* isolates, the community also contains four other Proteobacteria: *P. nigrifaciens* CCS176, *Halomonas sp900163645* CCS158, *V. casei* JB196, and *Hafnia alvei* JB232; two Firmicutes, *Staphylococcus xylosus* CCS177 and *S. carnosus* CCS178; four Actinobacteria, *Brevibacterium linens* JB5, *Brachybacterium* sp. CCS183, *Glutamicibacter arilaitensis* CCS165, and *Microbacterium gubbeenense* CCS160; and two fungi, *Debaryomyces hansenii* CCS145 and *Galactomyces geotrichum* CCS187. Isolates *V. casei* JB196 and *H. alvei* JB232 were chosen from our existing strain collection because these two species were represented in our genome catalog but not isolated from the communities as part of our isolation efforts. These two strains, together with the chosen *B. linens* JB5, were isolated as part of previous studies (5, 6) and are regularly utilized in our community experiments. The selected cheese rind isolates were then combined into *in vitro* communities following established protocols. In brief, a total of 100,000 CFU per community member were inoculated on the surface of a 10% Cheese Curd Agar petri dish (12). To mimic the production process for washed-rind cheeses, the surfaces of the plates were washed with 20% NaCl using a sterile swab every 24 h during the first 96 h, for a total of four washes. Communities were incubated in the dark at 15°C in a humidified environment (Fig. 6A). To assess the reproducibility of the *in vitro* washed cheese rind model, CFU counts of each sampling day (days 3, 5, 7, and 21) were performed for both the bacterial and fungal members (Fig. S10A, Table S18). In addition, a portion of each sample from sampling days 7 and 21 was used for DNA extraction and short-read metagenomic sequencing. Sequencing reads were then mapped back to reference genomes of the bacterial community members to track their relative abundance (Fig. 6B, Fig. S10B, Table S19). The additional sequencing and mapping allowed for analysis of *Psychrobacter* at the species level and revealed that all four species persisted within the full community. However, the relative proportion of each species varied over time, with *Psychrobacter* sp. CCS169 being most abundant on day 7 and the other three strains, especially *P. alimentarius* CCS164, increasing in relative abundance on day 21 (Fig. 6B, Fig. S10B).

Previous studies revealed a positive correlation between *G. geotrichum* and gammaproteobacterial species, and a negative correlation between *G. candidum* and actinobacterial species based on co-occurrence patterns in a sequencing-based survey of cheese rind microbiomes (5). In addition, pairwise experimental data showed stimulatory effects of *G. geotrichum* on gammaproteobacterial growth and inhibitory effects on actinobacterial growth (5). We took advantage of the fact that this model contains all three of these members (*G. geotrichum*, Proteobacteria, and Actinobacteria) to examine whether these patterns hold in a community context. To do this, we reconstructed a community lacking *G. geotrichum* and evaluated whether this resulted in any differences in the final community composition. The removal of *G. geotrichum* favored the relative growth of the Actinobacteria over the Proteobacteria (Fig. 6B, Fig. S10B, Table S19), even though overall bacterial absolute abundance was very similar between these two samples (Fig. S10A, Table S18). Absolute abundance based on read counts reveals that the decrease in gamma-proteobacterial abundance was largely due to the poor growth of *Pseudoalteromonas* (see Table S19). In contrast, almost all actinobacterial species reached higher absolute abundances compared to the full community. Overall, these results are consistent with *G. geotrichum* inhibiting Actinobacteria and stimulating Proteobacteria.

**FIG 5** Legend (Continued)
based on alignment of single-copy core genes. (B) Functional enrichment of COG categories in *Psychrobacter* genomes from cheese relative to other environments. (C) Pangenomic analysis of 8 *Psychrobacter* genomes from Cheese B. (D) Functional categories of 'core' (present in all 8 genomes), 'shell' (in 2 to 7 genomes), and 'cloud' (unique to 1 genome) gene clusters from 8 *Psychrobacter* genomes from Cheese B.

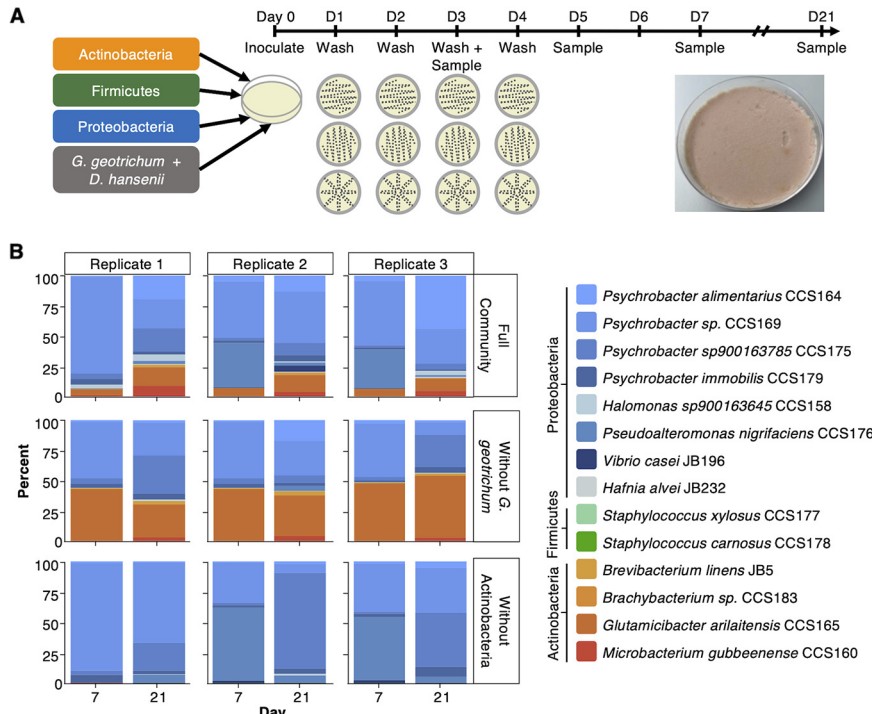

**FIG 6** A 16-member *in vitro* model system based on the washed-rind cheese microbiome. (A) Overview of the construction of washed-rind cheese-based *in vitro* model communities. An example plate of the full community on day 21 (D21) from the partially destructive sampling is shown. (B) Relative abundance plots of the *in vitro* communities after 7 and 21 days of growth based on short-read mapping to reference genomes.

We next tested the effect of removing all actinobacterial members from the community. Similar to the other conditions, *Psychrobacter* species dominated the communities, but increased to 93-95% in the communities without Actinobacteria. Overall, these communities cluster separately from the full community and the community without *G. geotrichum* in a principal-component analysis (Fig. S10C). Last, in contrast to the other communities, *P. alimentarius* CCS164 does not increase in relative abundance, remaining below 5% of *Psychrobacter* in all replicates (see Table S19). Instead, *Psychrobacter* sp. CCS169 competes with *Psychrobacter* sp. CCS175 for the most abundant *Psychrobacter*.

## DISCUSSION

Washed-rind cheeses harbor unique, moderately complex microbiomes which may provide useful systems for the study of microbial succession, interactions, strain-level dynamics, and HGT. This work represents a comprehensive, in-depth analysis of a set of washed-rind cheese microbiomes, combining culture-independent and culture-dependent approaches. Using amplicon and metagenomic sequencing, we showed that the three washed-rind cheeses which were the subject of this study showed reproducible succession patterns. Together, the amplicon sequencing (Fig. 1B to D) and long-read taxonomic classification (Fig. 2, Fig. S3B) showed that the mature communities of these cheeses were dominated by bacteria, in particular Proteobacteria, and indicated overall similar bacterial succession dynamics: both showed that Cheese A experienced relatively little taxonomic turnover between weeks 2 and 13 and that this community was consistently dominated by *Psychrobacter*, *Pseudoalteromonas*, and other Proteobacteria. In contrast, Cheese B showed more of a successional turnover, with *Psychrobacter* and *Pseudoaltermonas* initially being of comparable abundance within the community until *Pseudoalteromonas* eventually declined in abundance, while *Psychrobacter* and *Halomonas* became the abundant community members. For Cheese C, the amplicon and long-read-based taxonomic

classifications again revealed the same trend. Both showed a gradual takeover of the community by *Halomonas* with a concomitant rise of Gram-positive taxa such as *Brevibacterium*. Notably, there were also differences regarding the results from the amplicon and long-read-based classification. While many taxa were detected above the threshold by both techniques, each also resulted in unique taxa. For example, *Glutamicibacter* was detected above the threshold by long-read sequencing, but not by amplicon sequencing. Similarly, while the Bacteroidetes *Sphingobacterium* was detected by long-read sequencing, the Bacteroidetes *Mesonia* and *Myroides* were detected by amplicon sequencing. Interestingly, all three genera are represented in our genome catalog (Fig. 3A, Table S9). These differences are likely caused by biases inherent to the different sequencing techniques, such as uneven amplification of the 16S target gene from different taxa, as well as differences in the databases used. While the amplicon data were analyzed using the Greengenes database (29), the long-read amplicon data were classified using the NCBI nt database. The fact that the genome catalog contains MAGs of those taxa detected by both classification techniques as well as MAGS of those taxa detected by either of the techniques indicate that both techniques capture part of the ground truth, and that one technique alone is not sufficient to fully characterize the taxonomic diversity of these communities. Considering fungi, the amplicon and long-read data again contained similarities and differences, likely due to the same causes as for the bacterial part of the communities. Although the amplicon data, by nature of the technique, did not reveal details about the relative abundances of fungi and bacteria in the communities, the long-read data showed that the communities were already dominated by bacteria in week 2. The only exception was Cheese B, which was dominated by *Debaryomyces* in week 2. Other Debaryomycetaceae, Saccharomycetales, and *Geotrichum* made up most of the rest of the community at this time point. In contrast, the amplicon data suggest that the fungal communities of Cheese B at that time point were dominated by *Dipodascus*, while *Debaryomyces* constituted a smaller, yet substantial, part of the fungal communities in Cheese B on week 2. Again, these differences are likely due to amplification biases in the amplicon sequencing as well as database differences.

Overall, our taxonomic classification results are similar to what has been found for other washed-rind cheeses. For example, the communities investigated here were dominated by bacteria, while fungi made up a much smaller proportion of the community by the end of ripening (21, 22). In addition, we detected very similar bacterial and fungal genera as previous studies investigating the composition of washed-rind cheeses. For example, some taxa previously identified in washed-rind cheeses which were also identified in this study include the Proteobacteria *Psychrobacter* and *Halomonas*, the Actinobacteria *Brevibacterium* and *Glutamicibacter* (15, 17, 21, 22), and the fungi *Debaryomyces*, *Geotrichum*, and *Fusarium* (21–23). However, our multi-platform metagenomics approach provides new insights into species- and strain-level diversity and dynamics.

We also provide a catalog of high-quality MAGs (and isolate genomes) which spans the bacterial phyla Firmicutes, Bacteroidetes, Actinobacteria, and Proteobacteria (Fig. 3A, Table S9). In fact, two of the three cheeses yielded representative genomes from all four of these phyla despite low proportional abundance. It is likely that the long-read sequencing helped not only with recovering these low-abundance genomes, but also with the resolution of species- and/or strain-level MAGs. For example, we were able to resolve several *Psychrobacter* genomes in the three cheeses which may have been difficult to resolve using short-read sequencing alone. To assess the completeness of our genomic catalog, we mapped short reads from the same communities back to the catalog (see Table S11). The percentage of reads mapped to the catalog at each time point gave an indication of how well the catalog represented the community at the respective time points. Indeed, several time points for Cheeses B and C achieved read mapping percentages over 90%, suggesting that the genomic catalog captures the community diversity well. In contrast, only 19% of the reads from week 2 of Cheese B mapped to the respective genomic catalog. However, this made sense given that at this time point, this community was dominated by fungi, which were not

included in our catalog. In future studies, the robustness of these genomic catalogs can be estimated by mapping reads from novel washed-rind cheese communities and determining to what degree our generated catalog is representative of these communities in general. A similar approach was followed for a previously generated genomic catalog of dairy microbes (30).

When comparing the high-quality MAGs (and isolate genomes) between the three cheeses, we found that they contained both unique genomes and highly similar genomes (Fig. 3B, Fig. S4, Table S10). This finding is not surprising: on the one hand, we expected the three cheeses to share community members because they are ripened in the same facility and handled by the same cellar employees. The communities thus likely experience cross-inoculation during ripening. On the other hand, the milk sources, milk processing techniques (pasteurized, nonpasteurized), wash schedules, and ripening times (see Table S1) differed between the cheeses. As such, it is unsurprising that each also contained unique community members.

Our genomic catalog joins the ranks of previously created genomic catalogs, such as the recently created catalog of reference genomes from the human gut microbiome (31), which combined MAGs and isolate genomes; the Skin Microbial Genome Collection, which also integrates isolate cultivation with metagenomics (32); and other dairy microbial gene catalogs, such as one catalog of 117 isolate genomes (30). These genomic catalogs can be used to develop testable hypotheses, design genetic tools and high-throughput screening experiments that require reference genomes, and provide reference genomes for additional omics approaches, such as proteomics. In this study, we utilized the bacterial genomic catalog to further investigate the biology of these microbiomes, including the potential for HGT and the enrichment of specific functions. First, we leveraged the long reads together with metaHi-C reads to associate putative MGEs with their hosts. A similar approach was used by Kent et al. (33), where Hi-C was used to associate MGEs with their bacterial hosts in the human gut microbiome and track putative HGT events in longitudinal samples. Many of the MGEs detected in our data sets were not binned with their putative hosts during the initial binning and only became associated in this additional analysis (Fig. 4). We were able to assign 15%, 6%, and 29% of extrachromosomal MGEs to MAG hosts and 46%, 32%, and 22% of extrachromosomal MGEs to non-MAG contigs for Cheeses A, B, and C, respectively (Fig. S8). Using this method, we also detected several putative HGT events, including of MGEs predicted to encode iron-uptake pathways (see Table S14). Previous examination of cheese bacterial genomes revealed a strong enrichment of iron-uptake pathways on horizontally transferred regions (28). Additionally, some of the elements identified in this study which contain iron-uptake pathways also contain genes related to phosphanate transport. This association between horizontally transferred iron uptake regions and phosphanate transport genes has also been previously observed in cheese (28). The changes in host association as suggested by metaHi-C in this study require further *in vitro* confirmation and follow-up to fully understand their exact nature and impact within the cheese microbiome.

Next, we investigated the striking diversity of *Psychrobacter* within the communities. Genus-level diversity has been observed in other microbial communities, including cheese rinds, and their study *in vitro* has proven valuable. For example, by studying closely related, frequently co-occuring species of *Staphylococcus*, the importance of biotic interactions with fungi was revealed (7). Because in the communities investigated here, *Psychrobacter* showed the greatest sub-genus diversity in both the genomic catalog and among the isolates from Cheese B, we first followed up on this diversity by leveraging the genomic catalog. Specifically, we investigated functional enrichment within cheese-associated *Psychrobacter* MAGs and isolate genomes compared to that in *Psychrobacter* genomes associated with other environments (Fig. 5B). This question was of particular interest because *Psychrobacter* species are found in a variety of environments, both free-living and host-associated, and they show habitat-dependent adaptations (34). We identified an enrichment of iron acquisition genes and type VI

secretion genes in cheese-associated *Psychrobacter*. Both of these functional enrichments are reasonable given what is known about the cheese rind environment. First, the cheese microbiome has evolved to thrive in iron-limiting conditions (7, 35, 36). Second, the dense microbial communities that make up cheese rind microbiomes may lead to contact-dependent microbial interactions. Type VI secretion systems enable the delivery of cargo proteins which modulate bacterial-bacterial and bacterial-eukaryotic interactions (37). As such, contact-dependent interaction machinery enriched in cheese-associated *Psychrobacter* may play a larger role in species interactions in the densely populated cheese microbiome as opposed to interactions relevant to free-living, marine *Psychrobacter*, for example. Future *in vitro* studies could aim to understand how type VI secretion-mediated dynamics contribute to community composition and function.

The comprehensive, longitudinal nature of this data set makes it a potentially valuable resource for future investigations into the biology of cheese microbiomes or the study of microbiome dynamics in general. For example, the approaches applied here could be used to interrogate other types of MGEs, such as integrative and conjugative elements and prophages. Additionally, one aspect of the long-read data which we did not explore in this paper is the fact that it can detect methylation patterns on DNA, which can help associate extrachromosomal elements with their hosts and could theoretically be used to identify HGT events (38). Indeed, a recent study used this approach to recover MAGs from a marine microbial consortium and identify HGT events, phage infection, and strain-level structural variation (39).

Finally, we showed that, as in previous work on cheese microbiomes (5, 6), washed-rind communities are amenable to *in vitro* reconstruction (Fig. 6). The *in vitro* system established here represents a higher level of diversity than previous models and includes species- and strain-level diversity. This model community will allow further investigation of potential intrinsic and extrinsic factors which drive microbial ecology within these communities. For example, surface moisture and pH have been shown to impact community composition when comparing natural, bloomy, and washed-rind cheeses, while salt concentration was found to be less of a predictor (5). Nevertheless, given high exposure to salt during the ripening of washed-rind cheeses, this factor, too, should be examined more closely in these communities. Because the three communities examined in this study are subjected to brine salting, the interplay between salt content and surface moisture is another aspect that should be investigated *in vitro*.

Overall, the pairing of an experimental microbiome system with a corresponding in-depth metagenomic data set should provide ample future opportunities for generating and testing hypotheses related to the many facets of biology within these microbiomes.

## MATERIALS AND METHODS

**Cheese rind sample collection and DNA extraction.** Three production batches of three different washed-rind cheeses were sampled over the course of ripening, starting from fresh cheese wheels. At weeks 1, 2, 3, 4, 9, and 13 of ripening, the cheesemakers overnight-shipped two wheels of each sampled batch. The cheeses were stored at 4°C for up to 48 h prior to their harvest. During the harvest, rind was scraped off each wheel using fresh razor blades according to the methods of Wolfe et al. (5). The rind samples were homogenized by gentle stirring with sterile pipette tips. Rice grain-sized portions of these samples were frozen in cryotubes at −80°C in phosphate-buffered saline (PBS) + 40% glycerol. Where applicable, 10 mg of each wheel was collected in sterile 2-mL tubes for cross-linking as part of ProxiMeta Hi-C library preparation (see below). Finally, 500 mg of each sample was set aside in sterile 1.5-mL tubes at 4°C for DNA extraction. DNA was extracted within 72 h using a phenol-chloroform extraction protocol. If DNA was extracted at a later time, the rind samples were stored at −80°C.

For the DNA extraction, the rind samples were first disrupted mechanically by grinding into powder in liquid nitrogen (https://lab.loman.net/2018/05/25/dna-extraction-book-chapter/#__RefHeading___Toc505877552) using a mortar and pestle. The rind powder was incubated for 1 h at 37°C in 7 mL modified Tris-lysis buffer (10 mM Tris-Cl [pH 8], 100 mM EDTA [pH 8], 1% SDS, 20 $\mu$g/mL RNase A, 20 mg/mL lysozyme). Next, 87.5 $\mu$L of Proteinase K (800 units/mL) (New England Biolabs, Ipswich, MA, USA) was added, and the samples were mixed by inversion and incubated at 50°C for 1 h. The samples were then subjected to two rounds of DNA extraction using equal volumes of phenol-chloroform isoamyl alcohol. The final aqueous phase was mixed with equal volumes of ice-cold isopropyl and a 0.1-volume of 3 M sodium acetate. The precipitated DNA was pelleted in 5-mL centrifuge tubes at 17,000 × *g* for 3 min. When the volume of the final aqueous phase was

greater than 5 mL, the supernatant was removed, and the remaining sample was added to the same centrifuge tubes to repeat the pelleting step. The final pellets were washed in 1 mL ice-cold 70% ethanol and air-dried for about 15 min. Finally, the DNA was resuspended in 500 $\mu$L UltraPure DNase/RNase-Free Distilled Water (Thermo Fisher Scientific, Waltham, MA, USA) and kept at room temperature overnight to allow the DNA to dissolve. DNA was then stored at $-20$℃ until further processing.

**16S and ITS amplicon sequencing and analysis.** For 16S and ITS sequencing of the community DNA samples, we followed the Illumina-supplied "16S Metagenomic sequencing Library Preparation" protocol (part no. 15044223 rev. B) and "Fungal Metagenomic Sequencing Demonstrated Protocol" (document no. 1000000064940 v01), respectively. The 16S-specific primers (16S forward: 5'-TCGTCGGCAGCGTCAG ATGTGTATAAGAGACAG-CCTACGGGNGGCWGCAG-3', 16S reverse: 5'-GTCTCGTGGGCTCGGAGATGTGTATA AGAGACAG-GACTACHVGGGTATCTAATCC-3') targeted the 16S V3 and V4 regions. The ITS-specific primers (ITS forward: 5'-TCGTCGGCAGCGTCAGATGTGTATAAGAGACAG-CTTGGTCATTTAGAGGAAGTAA-3', ITS reverse: 5'-GTCTCGTGGGCTCGGAGATGTGTATAAGAGACAG-GCTGCGTTCTTCATCGATGC-3') were the same as the ITS_fwd_1 and ITS_rev_1 primers listed in the "Fungal Metagenomic Sequencing Demonstrated Protocol." For each of the two amplicon PCRs, 1.25 $\mu$L DNA (5 ng/$\mu$L) from the duplicate wheels for each sample was mixed and the resulting 2.5 $\mu$L was amplified with either the 16S- or ITS-specific primers and Q5 Hot Start High-Fidelity DNA polymerase (New England Biolabs). The PCR settings used were as follows: initial denaturation at 98℃ for 3 min; 25 rounds of 98℃ for 10 s, 55℃ for 30 s, and 72℃ for 15 s; and final extension at 72℃ for 2 min. Amplicon PCRs were purified using AMPure XP beads (Beckman Coulter, Indianapolis, IN, USA) and eluted with UltraPure DNase/RNase-Free Distilled Water (Thermo Fisher Scientific). For the index PCR, the 16S and ITS amplicon PCRs were subjected to amplification with IDT for Illumina Nextera DNA Unique Dual Indexes (set A, now called IDT for Illumina DNA/RNA UD Indexes) (produced by Integrated DNA Technologies [Coralville, IA, USA], sold by Illumina, Inc. [San Diego, CA, USA]) and Q5 Hot Start High-Fidelity DNA polymerase (New England Biolabs). Index PCRs were again purified using AMPure XP beads (Beckman Coulter, Indianapolis, IN, USA) and eluted with UltraPure DNase/RNase-Free Distilled Water (Thermo Fisher Scientific). The final index PCRs were quantified using the Qubit dsDNA HS kit (Thermo Fisher Scientific) with the Qubit 2.0 fluorometer (Thermo Fisher Scientific), diluted to 2 nM, and pooled at equimolar ratios. The pools were diluted to a final loading concentration of 50 pM and spiked with 50 pM PhiX control v3 (Illumina, Inc.) to a final concentration of 10% PhiX. Finally, 20-$\mu$L volumes of the 16S($+$PhiX) and ITS($+$PhiX) pools were sequenced individually in-house on an iSeq 100 (paired-end, 150 bp) (Illumina, Inc.).

The fastq files of the forward reads of the respective sequencing runs were imported into Qiime2 (version 2020.11.1) (40) and denoised with Dada2 (40–42) (qiime dada2 denoise-single -i-demultiplexed-seqs with flags -p-trim-left 0 -p-trunc-len 0 -p-n-threads 16). The denoised forward ITS reads were classified using classify-sklearn with a Naive Bayes classifier (qiime feature-classifier fit-classifier-naive-bayes) trained on a custom ITS database (5, 40, 41, 43–45). The denoised forward 16S reads were classified in the same way using a pretrained Greengenes database (Greengenes 13_8 99% operational taxonomic unit full-length sequences [MD5: 03078d15b265f3d2d73ce97661e370b1]) (29, 40, 41, 43–46). Detailed read statistics can be found in Tables S3 and S4. Ordination plots (principal coordinate analysis method, Bray-Curtis distance) were generated using R (version 4.0.2) (https://www.r-project.org/) together with the qiime2R (version 0.99.4), tidyverse (version 1.3.1), and phyloseq (version 1.34.0) packages (47–49). The stacked bar plots were also created with R using the qiime2R, tidyverse, and phyloseq packages. Taxonomies were collapsed at the genus level for both the ITS and 16S data. For the stacked bar plots showing a higher taxonomic level, taxonomies were collapsed at the phylum (16S) or order (ITS) level. In all cases, taxonomies with less than 1% abundance were aggregated under the category "Rest."

**metaHi-C.** For meta Hi-C, 10 to 15 mg of rind from duplicate wheels for each sample was fixed separately and then combined for further processing according to the ProxiMeta kit methodology from Phase Genomics (Seattle, WA, USA). The multiplexed libraries were sequenced by Novogene (Sacramento, CA, USA) on a HiSeq 4000 with a run configuration of 2 $\times$ 150 bp.

**Metagenomic shotgun sequencing.** For the HiFi sequencing, DNA from duplicate cheese wheels of each sample was combined in equal ratios. Five $\mu$g DNA of each sample was used as input for the library preparation, which was carried out by SNPsaurus (Eugene, OR, USA). The DNA was sheared to a modal size of 10 kb using a Megaruptor 2 (Diagenode Diagnostics, Seraing [Ougrée], Belgium). Libraries were prepared using Pacific Bioscience's Express Template Prep kit version 2.0 according to the manufacturer's protocol (https://www.pacb.com/wp-content/uploads/Procedure-Checklist-%E2%80%93-Preparing-Multiplexed-Microbial-Libraries-Using-SMRTbell-Express-Template-Prep-Kit-2.0.pdf; Pacific Bioscience, Menlo Park, CA, USA). Samples were pooled two at a time and size-selected using a BluePippin (Sage Sciences, Beverly, MA, USA) with the 0.75% Agarose Dye-free 10- to 18-kb cassette, U1 marker, and a 10-kb$+$ cutoff. The final libraries were sequenced by the Genomics and Cell Characterization Core Facility (University of Oregon, Eugene, OR, USA) on a Sequel II according to the SMRT Link Set Up (SMRT cell type = 8 M, sequencing kit = v2.0, sequencing primer = v2, binding kit = v2.0, sequencing control = v1, polymerase binding time = 4 h, movie time = 30 h, pre-extension time = 2 h, loading concentration = 100, 133, 150, or 200 pM [varied by cell], loading method = diffusion). HiFi reads were generated using the CCS tool (https://ccs.how/) version 6.2.0 with the default settings (ccs in.subreads.bam out.subreads.bam\–min-passes 3\–min-snr 2.5\–min-length 10\–max-length 50000\–min-rq \$MIN_RQ; \$MIN_RQ is set accordingly: ccs.Q20 - \$MIN_RQ = 0.99, ccs.Q30 - \$MIN_RQ = 0.999, ccs.Q40 - \$MIN_RQ = 0.9999).

Short-read sequencing libraries were prepared using the Nextera DNA Flex Library Prep workflow (Illumina, Inc.). For each sample, input DNA consisted of DNA from two duplicate cheese wheels from the same sample that was mixed in equal proportions (ng). For multiplexing, IDT for Illumina Nextera DNA Unique Dual Indexes (set A, now called IDT for Illumina DNA/RNA UD Indexes) (produced by Integrated DNA Technologies, sold by Illumina, Inc.) were used. Pooled libraries were sequenced by the

IGM Genomics Center at the University of California San Diego on the NovaSeq 6000 System using both lanes of a NovaSeq SP flow cell and a run configuration of 2 × 250 bp.

**Long-read based relative abundance estimations.** To investigate successional dynamics of the cheese communities based on HiFi reads, we utilized the Pacific Biosciences supplied toolkit "PB-meta-genomics-tools" (https://github.com/PacificBiosciences/pb-metagenomics-tools). First, HiFi reads were taxonomically classified with the "Taxonomic-Profiling-Nucleotide" pipeline, which uses minimap2 (50) to align sequences to the NCBI nt database and MEGAN-LR (51) to interpret alignments and assign reads to taxa. The read counts by taxonomy were generated using the "MEGAN-RMA summary." Outputs from the "MEGAN-RMA summary" were further processed in R (version 4.0.2) (https://www.r-project.org/) to-gether with the janitor (version 2.1.0), tidyverse (version 1.3.1), and cowplot (version 1.1.1) packages to generate the stacked bar plot shown in Fig. 2. Additionally, the outputs from the "MEGAN-RMA sum-mary" were used as inputs for the "compare-kreport-taxonomic-profiles" script available in the PB-meta-genomics-tools suite to generate Fig. S3.

**Generation of metagenome-assembled genomes.** Metagenomic assemblies were performed with hifiasm-meta (v. 0.2-r053) (52) using the default settings. For the individual time point assemblies, reads were input separately by time point and cheese. For the co-assemblies, reads from all time points were combined by cheese. To evaluate the assemblies and identify high-quality MAGs, we used the PacBio HiFi-MAG-Pipeline (https://github.com/PacificBiosciences/pb-metagenomics-tools, part of the "PB-meta-genomics-tools" suite). This pipeline uses minimap2 (50) to align HiFi reads to the contigs to obtain cov-erage estimates, which are used with MetaBat2 (53) to perform binning using all contigs. A separate bin set is also constructed from all circular contigs (e.g., one bin per circular contig), and the two binning strategies are compared and merged using DAS_Tool (54). The dereplicated bins are evaluated using CheckM (55), and quality thresholds are applied to retain high-quality MAGs (defaults: >70% complete-ness, <10% contamination, <20 contigs). In the final step of the pipeline, the high-quality MAGs are then analyzed using the Genome Taxonomy Database Toolkit (24), which attempts to identify the clos-est reference genome and assign taxonomy for each MAG.

**Isolation of bacterial community members.** The glycerol stocks of one wheel of Cheese B at weeks 2 and 13 were thawed on ice, homogenized by vortexing and pipetting, and aliquoted into smaller working stocks, which were frozen at −80°C for at least 24 h before further processing. A dilution series of one working stock each from weeks 2 and 13 was plated on plate count agar supplemented with milk and salt (PCAMS, 5 g/L tryptone, 2.5 g/L yeast extract, 1g/L dextrose, 1 g/L whole milk powder, 10 g/L so-dium chloride, and 15 g/L agar) containing 100 $\mu$g/mL cycloheximide to select against the fungal com-munity members. The plates were kept in a 15°C incubator for 4 days and then in the light at room tem-perature for an additional 3 days to allow for pigment formation. Colonies with as many distinct colony morphologies as could be distinguished by eye were then purified by re-streaking three times on plain PCAMS, each time with an incubation at 15°C for 4 days and an additional 3-day incubation in the light at room temperature. Colonies from the final re-streak were patched onto plain PCAMS and incubated at 15°C for 4 days. Two-mL overnight cultures (LB Miller broth) were inoculated from the patches and grown shaking at 240 rpm at 22°C. After 24 h, glycerol stocks were prepared using PBS + 20% glycerol, flash-frozen, and stored at −80°C. The only exception to this was CCS196 (*Psychrobacter* sp.), which was grown on PCAMS agar at 22°C for 24 h, harvested in PBS + 20% glycerol, flash-frozen, and stored at −80°C.

**Short-read, whole-genome sequencing of Cheese B isolates.** Isolates (CCS156, CCS158, CCS160, CCS164, CCS165, CCS166, CCS169, CCS174, CCS175, CCS176, CCS177, CCS178, CCS179, CCS180, CCS181, CCS182, CCS183, CCS184, CCS196) were grown from glycerol stocks on plain PCAMS for 2 days at 15°C and an additional 3 (all except CCS183) or 5 days (only CCS183) at room temperature on the benchtop. DNA was extracted following the Nextera DNA Flex Microbial Colony Extraction protocol (document no. 1000000035294v01) with the following modifications: AMPure XP beads (Beckman Coulter, Indianapolis, IN, USA) were used instead of SPB beads, and 2-mL cryotubes filled with approximately 250 $\mu$L acid-washed beads (1:1 ratio of 425- to 600-$\mu$m and 150- to 212-$\mu$m beads) were used instead of PowerBead Tubes. In addition, cells were collected from the primary streak using a sterile pipette tip. The extracted DNA was quantified using the Qubit dsDNA HS kit (Thermo Fisher Scientific) with the Qubit 2.0 fluorom-eter (Thermo Fisher Scientific) and stored at −20°C. Steps 25 to 27 were skipped. Due to small amounts of cellular input, steps 7 and 8 were skipped for CCS183, and instead 50 $\mu$L was mixed with 20 $\mu$L AMPure beads. For the library preparation, a maximum of 500 ng of DNA was mixed with UltraPure DNase/RNase-Free Distilled Water (Thermo Fisher Scientific) for a total volume of 30 $\mu$L. The libraries were prepared using the Illumina DNA Prep protocol (document no. 1000000025416 v09) and the IDT for Illumina DNA/RNA UD Indexes (set A; produced by Integrated DNA Technologies, sold by Illumina, Inc.). Tagmented DNA was amplified for 5 cycles, except in the case of CCS191, which was amplified for 8 cycles due to low input amounts. Libraries were diluted to 30 nM using buffer RSB from the Illumina DNA Prep protocol, pooled at equimolar ratios, and sequenced by Novogene on a HiSeq 4000 with a run configuration of 2 × 150 bp.

**Long-read whole-genome sequencing of Cheese B isolates.** A subset of isolates (CCS156, CCS158, CCS160, CCS164, CCS165, CCS166, CCS169, CCS174, CCS176, CCS179, CCS180, CCS181, CCS182, CCS183, CCS184, and CCS196) was additionally sequenced using Nanopore technology (Oxford Nanopore Technologies, Oxford, United Kingdom) to improve isolate assemblies. Isolates were grown either shak-ing at room temperature in 2 mL LB Miller broth for at least 24 h until the culture appeared cloudy (CCS156, CCS158, CCS160, CCS164, CCS165, CCS166, CCS169, CCS174, CCS176, CCS179, CCS180, CCS182, CCS183, CCS184) or on PCAMS agar plates at room temperature for ~48 h (CCS181 and CCS196). Cells from liquid cultures were harvested by centrifugation at 10,000 rpm for 5 min, removal of supernatant by pipetting, and freezing at −80°C. Cells from the agar plates were recovered by adding 2.5 mL PBS

onto the plate and dislodging cells with sterile cell scrapers. One mL of recovered cell suspension was centrifuged at 10,000 rpm for 5 min and frozen at −80°C after supernatant removal. Cell pellets were thawed, and DNA was extracted using the phenol-chloroform protocol described above without the liquid nitrogen grinding step (CCS156, CCS165, CCS166, and CCS176) or using the Qiagen Genomic-tip 20/G kit (Qiagen, Venlo, Netherlands) according to the manufacturer's instructions. DNA of CCS156, CCS165, CCS166, and CCS184 was extracted from the pellet resulting from the full 2-mL culture, while DNA from the rest of the isolates was extracted from pellets resulting from 1 mL of the overnight culture. DNA extractions were quantified using the Qubit dsDNA HS kit (Thermo Fisher Scientific) with a Qubit 2.0 fluorometer (Thermo Fisher Scientific) and their quality was assessed with the Tapestation gDNA assay (Agilent, Santa Clara, CA, USA). DNA was stored at –20°C until the library preparation was carried out using the Nanopore kit SQK-LSK110 (the DNA control strand was included in the library preparation for CCS156, CCS165, CCS166, and CCS176, but not for the others). Libraries were either sequenced right away using a flongle flow cell (FLO-FLG001) or stored at –80°C until sequencing. For most samples, basecalling was done in real-time during the sequencing run (CCS156, CCS158, CCS160, CCS164, CCS166, CCS169, CCS174, CCS179, CCS180, CCS181, CCS182, CCS183, CCS184, and CCS196) using the fast basecalling model in the MinKnow software (version 21.02.1 running Guppy version 4.3.4 for CCS156, CCS158, CCS166, CCS180, CCS181, CCS183 and CCS196 or version 21.06.0 running Guppy version 5.0.11 for CCS160, CCS164, CCS169, CCS174, CCS179, CCS182 and CCS184) with a minimum q-score of 7. For a couple of samples (CCS165 and CCS176), basecalling was done post-sequencing using Guppy (version 5.0.16) on the command line with the fast basecalling model (config file dna_r9.4.1_450bps_fast.cfg) and the minimum q-score set to 7. We recovered read N50s between 4 and 30 kb. In all cases, the fastq files in the "pass" folders were concatenated by isolate and used for further analysis.

**Isolate genome assemblies.** To generate the isolate assemblies, the Illumina reads were first trimmed using Trimmomatic (56) (version 0.39) using the paired-end mode and the arguments ILLUMINACLIP:NexteraPE-PE.fa:2:40:15:2:True SLIDINGWINDOW:4:20 MINLEN:20. The paired and unpaired trimmed reads of CCS175, CCS177, and CCS178 were assembled using SPAdes (57) (version 3.13.0, with arguments -t 14 -m 50 -k 33,55,77,99,127). For CCS156, CCS158, CCS160, CCS164, CCS165, CCS166, CCS169, CCS174, CCS176, CCS179, CCS180, CCS181, CCS182, CCS183, CCS184, and CCS196, hybrid assemblies were performed with SPAdes (58) using both the paired and unpaired trimmed Illumina reads and the Nanopore reads as input (version v3.13.0 with arguments -t 14 -m 50 -k 33,55,77,99,127). Contigs below 1,000 bp were removed.

**Generation of genomic catalog.** MAGs resulting from co-assemblies and individual time point assemblies (and isolate assemblies in case of Cheese B) were dereplicated with dRep (version 3.2.2) (59). CheckM (version 1.1.3) (55) results were provided as an Excel sheet. Genomes for the genomic catalog were selected by hand from the resulting MASH clusters by prioritizing isolates (for Cheese B) over circular individual time point MAGs over circular co-assembly MAGs over complete, noncircular individual time point MAGs over complete, noncircular co-assembly MAGs. If several genomes per cluster fell into the same category, the genomes were prioritized based on low contamination and high completeness as determined by CheckM. Taxonomies of the final genomes in the genomic catalog were determined by GTDB-Tk (version 1.6.0, classify workflow) (24). To determine the abundance of the genomes from the genomic catalog over time, the genomes in the catalog were indexed using bwa index and the Illumina reads from weeks 2 to 13 were mapped to this indexed genomic catalog using bwa mem (version 0.7.17-r1188) (60). Before mapping, the short-read metagenomic reads were concatenated by sample across the two lanes of the sequencing run and trimmed using Trimmomatic (56) (version 0.39) using the paired-end mode and the arguments ILLUMINACLIP:NexteraPE-PE.fa:2:40:15:2:True SLIDINGWINDOW:4:20 MINLEN:20. Samtools (61) (version 1.9 using htslib 1.9) was used to convert .sam files into .bam files with samtools view, and .bam files from all time points of each cheese were combined using samtools cat. Samtools flagstat was used to determine the percentage of reads mapped to the genomic catalog. Anvi'o (version 7.1 'hope') (62) was used to generate the plots in Fig. 3A.

**Generation of mega-assemblies.** For each cheese, contigs from the co- and individual time point assemblies were concatenated, and contigs of <1,000 bp were removed. The concatenated assemblies were indexed using bwa index, and short-read metagenomic reads were mapped to this indexed, concatenated assembly using bwa mem (version 0.7.17-r1188) (60). Before mapping, the short-read metagenomic reads were concatenated by sample across the two lanes of the sequencing run and trimmed using Trimmomatic (56) (version 0.39) using the paired-end mode and the arguments ILLUMINACLIP:NexteraPE-PE.fa:2:40:15:2:True SLIDINGWINDOW:4:20 MINLEN:20. Samtools (61) (version 1.9 using htslib 1.9) was used to convert .sam files into .bam files with samtools view, and .bam files from all time points of each cheese were combined using samtools cat. Samtools view was then used to extract all short-read metagenomic reads that did not map to the indexed, concatenated assemblies. The resulting .bam files were converted into fastq files using the bamToFastq command (version 0.5.3) from the BEDTools suite (63), and fastq_pair from the fastq-pair package (version 1.0) (64) was used to synchronize the newly generated fastq files. The reads in the resulting fastq files were assembled using SPAdes (58) using both the paired and unpaired trimmed Illumina reads as input (version v3.13.0 with arguments -t 14 -m 50 -k 33,55,77,99,127). To generate the mega-assembly for each cheese, unbinned contigs from the co- and individual time point assemblies of the respective cheese were first dereplicated. To dereplicate these contigs, a custom nucleotide blast database was created consisting of the unbinned contigs. Unbinned contigs were then compared to this database using blastn (65, 66) (version 2.10.1+, options: -outfmt "6 qseqid sseqid pident qcovs length qlen slen evalue score" -evalue 1e-6 -perc_identity 99 -word_size 20 -num_threads 12). A contig was considered redundant if there was an overlap of contigs of at least 90%; in this case, the smaller contig was considered redundant. Seqkit (67) (version 2.2.0) was then used to extract the nonredundant unbinned contigs from the full set of unbinned contigs. These

nonredundant unbinned contigs were then combined with the contigs resulting from the assembly of the unmapped Illumina reads and the contigs from the selected dereplicated MAGs in the genomic catalog. The final mega-assembly for each cheese should incorporate all nonredundant assembly information captured from the combination of Illumina and PacBio sequencing.

**Identification of mobile genetic elements.** Plasmid contigs were identified in each cheese metagenome by analyzing the mega-assemblies (contigs of >5,000 bp) with viralVerify (https://github.com/ablab/viralVerify, version 1.1) using the -p flag and Pfam database version Pfam33.1. Viruses were identified with VIBRANT (68) (version 1.2.1). Numbers of predicted plasmid and virus contigs were plotted using R (version 4.0.2) (https://www.r-project.org/) and the readxl (version 1.3.1) and ggplot2 (version 3.3.5) packages.

**Assigning MGEs to hosts.** To assign MGEs to hosts, we used the viralAssociationPipeline.py script (https://github.com/njdbickhart/RumenLongReadASM/blob/master/viralAssociationPipeline.py) (26, 27). In brief, the contigs in the mega-assemblies for each cheese were classified taxonomically using Kraken2 and a custom Kraken database containing the default Kraken database supplemented with genomes of cheese-associated microbes. The outputs were reformatted in Excel to fit the input requirements for the viralAssociationPipeline.py script. MetaHi-C reads were then aligned to the indexed mega-assemblies using bwa mem (version 0.7.17-r1188) (60) (bwa mem -v 1 -t 16 -5SP {mega-assembly} {forward_Hi-C} {reverse_Hi-C}). Reads which mapped to multiple locations in the assemblies were removed using grep -v -e 'XA:Z:' -e 'SA:Z:'. The resulting .sam files were then converted into .bam files using samtools view (61). seqtk subseq (https://github.com/lh3/seqtk) was used to extract the predicted MGE contigs as a .fasta file. The lengths of the contigs in this file were determined using samtools faidx (61) (version 1.9 using htslib 1.9). The viralAssociationPipeline was then run using the -a, -g, -b, -i, -v, -o, -s, -m, and -l flags. iTOL (69) was used to generate the diagram in Fig. 4 showing the connections between MAGs (and isolate genomes in the case of Cheese B) and MGEs (plasmids as predicted by viral verify -p, lytic viruses as predicted by Vibrant). A summary graph showing the numbers of extrachromosomal MGEs associated with MAGs, with unbinned contigs and the numbers of unassociated extrachromosomal MGEs (Fig. S8), was plotted using R (version 4.0.2) (https://www.r-project.org/) and the readxl (version 1.3.1) and ggplot2 (version 3.3.5) packages.

***Psychrobacter* pangenome analysis.** To identify a nonredundant set of *Psychrobacter* genomes from our cheese isolate assemblies and the full set of MAGs from the three cheeses, genomes classified by GTDB-Tk (version 1.6.0, classify workflow) (24) as belonging to *Psychrobacter* were dereplicated with dRep (version 3.2.2) (59). CheckM (version 1.1.3) (55) results were provided as an Excel sheet. Genomes were selected by hand from the resulting MASH clusters by prioritizing isolates over circular individual time point MAGs over circular co-assembly MAGs over complete, noncircular individual time point MAGs over complete, noncircular co-assembly MAGs. This resulted in the selection of 2 MAGs (Cheese B MAG 21 and Cheese A MAG 15) and 7 isolate genomes. An additional 97 publicly available *Psychrobacter* genomes were downloaded from NCBI (https://www.ncbi.nlm.nih.gov/assembly/) or from the JGI Integrated Microbial Genomes and Microbiomes system (see Table S15). The full data set consisted of 17 genomes from cheese, 8 from other fermented foods, 35 host-associated, 10 from soil, 23 from marine environments, and 13 from other miscellaneous environments. These 106 genomes were analyzed using the microbial pangenomics workflow in Anvi'o (62, 70) (version 7.1 'hope'). Specifically, assemblies were run through gene prediction with Prodigal (71) (version 2.6.3), hits to bacterial single-copy gene collections were identified using HMMER (http://hmmer.org/), and genes were annotated with functions from the NCBI's Clusters of Orthologous Groups (72). Identification of core and accessory gene sets and the construction of the phylogenetic tree based on single-copy core gene SNPs was done by panX (73) (version 1.6.0) using FastTree 2 (74) and RaxML (75). The core gene set was defined here as being present in at least 95% of genomes. Functional enrichment analysis was performed in Anvi'o with a comparison of genomes from cheese versus those not from cheese, a corrected q-value cutoff of 0.1, and COG20_FUNCTION as the annotation source (76). The separate pangenomic analysis of the 8 *Psychrobacter* genomes from Cheese B was performed as described for the full data set, and the anvi-summarize function in Anvi'o was used to determine the core, shell, and cloud gene sets and associated COG functional categories. The summary chart was produced based on the Anvi'o output using Microsoft Excel for Mac version 16.53.

***In vitro* washed-rind community reconstitution.** For the *in vitro* community reconstitution, 16 cheese isolates were chosen based on metagenomic analyses. From this study, four *Psychrobacter* strains (CCS164, CCS169, CCS175 and CCS179), one *Halomonas* strain (CCS158), one *Pseudoalteromonas* strain (CCS176), one *Glutamicibacter* strain (CCS165), two *Staphylococcus* strains (CCS177, CCS178), one *Brachybacterium* strain (CCS183), one *Microbacterium* strain (CCS160), and two fungal strains (D. hansenii CCS145 and G. geotrichum CCS187) were chosen. In addition, three strains from a previous study were included in the community (B. linens JB5, V. casei JB196, and H. alvei JB232) (5, 6). Strains of the 16-member community were inoculated at equal ratios (at 100,000 bacterial cells or fungal spores each) on 10% cheese curd agar (12). For each community and sampling condition, triplicate communities were inoculated. Agar plates were incubated in the dark at 15°C in a humidified plastic bag. At 24, 48, 72, and 96 h after inoculation, plates were scrubbed with a 20%-wt NaCl solution using sterilized cotton swabs in a horizontal and vertical rastering pattern, followed by a rosette (Fig. 6A). On days 3, 5, 7, and 21, microbial communities were collected into 1,000 $\mu$L of PBS + 0.05% Tween using cell scrapers; for the day 3 sample, biomass collection was done before the brine wash. Half of each sample was split for various analyses: spot plating for CFU determination, glycerol stock preparation, and DNA extraction for metagenomic short-read sequencing in the case of days 7 and 21. Half of the scraped biomass was then replated onto the same petri dish. To calculate total bacterial and fungal CFU, spot plating of serial dilutions of each sample was done on PCAMS, either with 50 $\mu$g/L chloramphenicol for fungal community members or 100 $\mu$g/L cycloheximide + 21.6 $\mu$g/L natamycin for bacterial community members, with colony counting performed 48 h later (see Table S18). DNA was extracted from the *in*

*vitro* community samples from days 7 and 21 using phenol-chloroform extraction (without a liquid nitrogen grinding step) and purified with 5 additional ethanol washes to remove residual phenol and chloroform. Library preparation and metagenomic sequencing of the multiplexed libraries were performed by Novogene on a NovaSeq 6000 with a run configuration of $2 \times 150$ bp.

Metagenomic reads were then mapped back to the bacterial reference genomes using bwa mem to show relative abundance. For *B. linens* (JB5, accession no. KF669529), *V. casei* (JB196, https://gold.jgi.doe.gov/analysis_projects?id=Ga0212130), and *H. alvei* (JB232, accession no. KF669544), previously published reference genomes were utilized. For the CCS strains, the *de novo* assembled genomes from this study were utilized. A total of 1,259,325,714 reads were sequenced, and over 99.7% of reads were aligned against the reference bacterial genomes (Table S19). Anvi'o (version 7.1 'hope') (62) was used to visualize the metagenome and the coverage from short reads. The coverage values of the contigs were filtered for nucleotide positions that were within the interquartile range (25% to 75%) of coverage values for each contig before being averaged. Across the contigs of each genome, these coverage values were also averaged, weighted by their length. To calculate the relative abundance for each community member, the coverage values were divided by the sum of all coverage values. The principal-component analysis (PCA) plots of the coverages were prepared in R (version 4.0.2) (https://www.r-project.org/) and the tidyverse (version 1.3.1) and ggfortify (version 0.4.14) (77) packages. The principal-component analysis was carried out using the prcomp function (center = TRUE, scale. = TRUE).

**Computational resources.** Computationally intensive analyses were run on the Triton Shared Computing Cluster (78).

**Data availability.** Fastq files of all metagenomic raw sequencing data are available on the Sequence Read Archive under accession no. PRJNA778418. Subread files of the HiFi sequencing are available upon demand. Fastq files of all isolate sequencing data are available on the Sequence Read Archive under accession no. PRJNA837750 (CCS156), PRJNA837754 (CCS158), PRJNA837770 (CCS160), PRJNA837776 (CCS164), PRJNA838264 (CCS165), PRJNA837782 (CCS166), PRJNA837789 (CCS169), PRJNA838091 (CCS174), PRJNA838092 (CCS175), PRJNA838093 (CCS176), PRJNA838094 (CCS177), PRJNA838095 (CCS178), PRJNA838105 (CCS179), PRJNA838104 (CCS180), PRJNA838102 (CCS181), PRJNA838100 (CCS182), PRJNA838106 (CCS183), PRJNA838262 (CCS184), and PRJNA838261 (CCS196). Fastq files of the *in vitro* community sequencing are available on the Sequence Read Archive under accession no. PRJNA852571. Assemblies, MAGs, and supplemental files (Tables S1-S19) have been deposited on Dryad (https://doi.org/10.5061/dryad.bg79cnpd8).

## SUPPLEMENTAL MATERIAL

Supplemental material is available online only.

**FIG S1**, PDF file, 0.3 MB.

**FIG S2**, PDF file, 0.2 MB.

**FIG S3**, PDF file, 0.8 MB.

**FIG S4**, PDF file, 0.1 MB.

**FIG S5**, PDF file, 0.1 MB.

**FIG S6**, PDF file, 0.1 MB.

**FIG S7**, PDF file, 0.02 MB.

**FIG S8**, PDF file, 0.1 MB.

**FIG S9**, PDF file, 0.1 MB.

**FIG S10**, PDF file, 0.3 MB.

## ACKNOWLEDGMENTS

We thank Jasper Hill Farm for supplying the cheese samples, Pacific Biosciences for supplying HiFi sequencing reagents and project support, and SNPSaurus and the Genomics and Cell Characterization Core Facility at the University of Oregon for providing HiFi sequencing services. In addition, we thank the Graduate Women in Science fellowship awarded to C.C.S. that allowed the generation of metaHi-C data, and the IGM Genomics Center at the University of California San Diego/Illumina for a NovaSeq sequencing grant to E.C.P. The work was additionally supported by funds from the National Institutes of Health (DP2 AT010401), the David and Lucile Packard Foundation (no. 2016-65131), and the Pew Charitable Trusts (Pew Scholar in Biomedical Sciences) to R.J.D.

We thank Steven Villareal for help with DNA extractions from Cheese B isolates for Nanopore sequencing, Ivan Liachko from Phase Genomics and Mathieu Almeida for advice on the bioinformatic analyses, Brooke Johnson for help with exploratory data analysis, Lucas Patel for help with the *in vitro* community experiment, and Benjamin Wolfe for helpful feedback on the manuscript.

C.C.S., R.J.D., M.A., and R.H. conceptualized the study. C.C.S., with help from E.C.P., collected the samples and generated the metagenomic sequencing data. C.C.S. isolated

community members from Cheese B and generated their corresponding sequencing data. C.B.D. conducted the *in vitro* community experiment. C.C.S. and E.C.P. analyzed the data. D.P. generated the long-based assemblies and MAGs. C.C.S. and E.C.P. prepared the figures. C.C.S. wrote the manuscript with input from all authors. All authors agree with the contents of the manuscript.

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
