## [Reviewer comments · mSystems]

Longitudinal, multi-platform metagenomics yields a high-quality genomic catalog and guides an *in vitro* model for cheese communities

Christina Saak, Emily Pierce, Cong Dinh, Daniel Portik, Richard Hall, Meredith Ashby, and Rachel Dutton

Corresponding Author(s): Rachel Dutton, University of California, San Diego

Review Timeline:

Submission Date:	July 30, 2022
Editorial Decision:	September 21, 2022
Revision Received:	November 12, 2022
Accepted:	November 22, 2022

Editor: Paul Cotter

Reviewer(s): Disclosure of reviewer identity is with reference to reviewer comments included in decision letter(s). The following individuals involved in review of your submission have agreed to reveal their identity: Davide Porcellato (Reviewer #1)

Transaction Report:

DOI: <https://doi.org/10.1128/msystems.00701-22>

September 20, 2022

Dr. Rachel J Dutton
University of California, San Diego
AP&M Building, room 3872
9500 Gilman Drive
La Jolla, CA 92093-0116

Re: mSystems00701-22 (Longitudinal, multi-platform metagenomics yields a high-quality genomic catalog and guides an *in vitro* model for cheese communities)

Dear Rachel:

Thank you for submitting your manuscript to mSystems. We have completed our review and I am pleased to inform you that, in principle, we expect to accept it for publication in mSystems. However, acceptance will not be final until you have adequately addressed the reviewer comments.

Ultimately, the two reviewers very much liked the work but both had issues in relation to the presentation (request for an expanded introduction, some more discussion re the implications of the study, comments relating to figures and supplemental data, clarity re why the specific focus on *Psychrobacter*). While these are not trivial (and one reviewer recommended rejection with a view to resubmission), I think it should be possible to address all of the points in a reasonable time frame

Preparing Revision Guidelines

Sincerely,

Paul

Editor, mSystems

Journals Department
Reviewer comments:

Reviewer #1 (Comments for the Author):

The manuscript by Saak et al. deals with the investigation of the cheese rind microbiome and its genomic potential and temporal dynamics. The work has been performed using the latest short- and long-sequencing technologies and bioinformatics tools. The work is interesting, and the amount of data and results generated is incredible. The different approaches (culture-independent, culture-dependent, in vitro washed rind cheese system, pangenome) provide a good understanding of this complex microbiome from different angles. The following comments/suggestions must be considered before the manuscript can be accepted.

The discussion should be improved and extended to discuss more potential intrinsic and extrinsic factors that could have driven the microbial ecology on the rind, as the authors suggest in the introduction. What has caused this microbiome to shift over time? For example, there is no mention of changes in salt concentration or pH during cheese ripening which, for sure, impact the rind microbiome. How was the cheese salted? Dry or brine salting? The salt diffuses into the cheese over time. Was this considered? pH changes during maturation are also important and need discussion.

I am also wondering if the pangenome analysis of the *Psychrobater* needs to be included in this work. The results are exciting and confirm what was already found by other results in the paper, but in my opinion, these data should be explored more and discussed further. The authors described this part as "an example," but I think these results have a more significant potential if exploited more. Can these data be presented in more detail in another paper?

As I said, much work has been done, and I understand the difficulties in presenting all the data in detail in one manuscript. However, some figures are too complex, and the visualization is difficult to follow-for example, figures 3A, 5A, and 5D.

Minor comments

Several references are given at the same point. I think a maximum of 3 references are enough for each statement. The introduction can be extended.

Line 88: Remove "genome". Plasmids are not associated with host genomes but with the host.

Line 575: remove "of washed-rind cheese communities". Not necessary here.

Line 618: Add the species name of the CCS196 if available

Line 370: is Fig 5A the correct number?

The author should give more information on the cheeses that were used for the study. For example, mention production technologies used and other parameters which would help understand changes in the rind microbiome as mentioned above (water content or dry matter of the fresh cheese, salt, pH, ripening temperature, humidity, starter culture used ...).

Was there any possibility of cross-contamination between the cheeses during ripening since they were ripened in the same facility?

Line 470. Cheese rind was stored for up to 72h before DNA extraction in addition to the 48 h of cheese storage. Could these two storage times impact the community composition and, therefore, functionality described after metagenomic analysis?

DNA extraction was done only with lysozyme and proteinase K. This introduces a bias. Did the authors check this? A combination of mechanical disruption and enzymatic treatment of the microbial pellet is suggested in microbiome studies. Alternative two different DNA extraction methods are a better alternative for the two different sequencing technologies (short and long DNA sequencing).

Reviewer #2 (Comments for the Author):

This paper is a complex study that incorporates various methods to understand succession dynamics in cheese rind

microbiomes at various levels of resolution. It could be improved with more discussion, as it is currently very result-heavy. Further discussion on the potential application of these methods, in particular, the benefits and uses/application of a genomic catalog, the comparisons with existing studies on the cheese rind microbiome and with other studies that have employed such methods would elevate the discussion. There are too many supplementary figures and tables that can be removed (supp table 8 and 13) or combined (supp figure 1-2, supp tables 9-11).

Some questions:

1. While the consistent successional patterns were evident, were there any differentially abundant taxa/genera (bacteria/fungal) between batches? (L29-31)
2. Was the long read data in Figure 2 and Supp figure 5 in agreement with the amplicon data in Figure 1? Were there any differences?
3. There were taxa that were specific to certain cheeses and some that were shared between all cheeses - have these been seen in other studies? Elaborate more on this in the discussion, with examples.
4. GTDB was used for MAGs - taxa differs in GTDB and NCBI databases - why wasn't NCBI used with MEGAN, which was used for long read taxonomic classification?
5. While Figure 3 is a very nice figure, it might be clearer if expressed in a table, like a summary table of the MAGs selected for the genomic catalog.
6. Not so clear what the purpose of Supp table 12 - if it is to just mention that several bins contain similar/identical genomes? Or that isolates and assembled MAGs are the same?
7. Supp table 13 - can just give a range in the text, don't need another table.
8. Genomic catalog - How should the robustness of the genomic catalog be evaluated? - are there other papers that specify this? (L220-222). It would also be nice to give examples of the use of this in future studies or cite papers that have done so.
9. Why was *Psychrobacter* chosen for pangenome analysis and what is the importance of the co-existence of multiple *Psychrobacter* spp.?
Also the categories of the other genomes included (supp table 17) were a little strange/inconsistent? (e.g. Marine sediment and lake water classified as misc. and not marine? Lake water as misc)
10. L366-382 and Supp table 19: these results might be better visualised as a figure, instead of Supp table 19. If they are already in figures, please refer to them. It also might be good to combine a few of the relevant ones into one supp figure.

Other more minor/specific comments:

- As there are many figures and supplementary figures and tables, it would be good if they were referred to in the text (e.g. L129-133: can add reference to Figure 1D, and similar for L133-138: refer to Figure 1C.) Also rearrange figures so that those mentioned first appear earlier - i.e. Figure 1C and D referred to before figure 1B, Supp Fig 3B referred to before Supp Fig 3A.
- L181: what about the other 5 from cheese A and 14 from cheese C?
- L223-241: might be good if it was visualised somehow? Also refer to previous figures in the text.
- L223-241: would be nice to show in a figure the species and strain-level dynamics from short reads mapped back to the genomic catalog
- L289-290: last part should be in the discussion and not the results. L359-363, these should be moved to the discussion as they are not results
- does the adjusted q-value indicate significance?
- L349, 351, 419: days 3, 5, 7, 21 - standardise L807-808: 3 strains from a previous study, were these not found in this study? Is there a reason they were added in?
- Standardise - Dereplicate or de-replicate?
- In general, the font size in the Figures could be increased, quite hard to read at 100%
- Supplementary tables - lines are messy, can be cleaned up and fonts can be standardised
- Figure 5B: might be good to underline the genomes from this study
- Supp figure 12 can be made an actual figure, not in supplementary
- Supp table 19: please check day 0 replicate 1 for both bacteria and fungi

This paper is a complex study that incorporates various methods to understand succession dynamics in cheese rind microbiomes at various levels of resolution. It could be improved with more discussion, as it is currently very result-heavy. Further discussion on the potential application of these methods, in particular, the benefits and uses/application of a genomic catalog, the comparisons with existing studies on the cheese rind microbiome and with other studies that have employed such methods would elevate the discussion. There are too many supplementary figures and tables that can be removed (supp table 8 and 13) or combined (supp figure 1-2, supp tables 9-11).

Some questions:

1. While the consistent successional patterns were evident, were there any differentially abundant taxa/genera (bacteria/fungal) between batches? (L29-31)
2. Was the long read data in Figure 2 and Supp figure 5 in agreement with the amplicon data in Figure 1? Were there any differences?
3. There were taxa that were specific to certain cheeses and some that were shared between all cheeses – have these been seen in other studies? Elaborate more on this in the discussion, with examples.
4. GTDB was used for MAGs – taxa differs in GTDB and NCBI databases – why wasn't NCBI used with MEGAN, which was used for long read taxonomic classification?
5. While Figure 3 is a very nice figure, it might be clearer if expressed in a table, like a summary table of the MAGs selected for the genomic catalog.
6. Not so clear what the purpose of Supp table 12 – if it is to just mention that several bins contain similar/identical genomes? Or that isolates and assembled MAGs are the same?
7. Supp table 13 – can just give a range in the text, don't need another table.
8. Genomic catalog – How should the robustness of the genomic catalog be evaluated? – are there other papers that specify this? (L220-222). It would also be nice to give examples of the use of this in future studies or cite papers that have done so.
9. Why was *Psychrobacter* chosen for pangenome analysis and what is the importance of the co-existence of multiple *Psychrobacter* spp.?

Also the categories of the other genomes included (supp table 17) were a little strange/inconsistent? (e.g. Marine sediment and lake water classified as misc. and not marine? Lake water as misc)

10. L366-382 and Supp table 19: these results might be better visualised as a figure, instead of Supp table 19. If they are already in figures, please refer to them. It also might be good to combine a few of the relevant ones into one supp figure.

Other more minor/specific comments:

- As there are many figures and supplementary figures and tables, it would be good if they were referred to in the text (e.g. L129-133: can add reference to Figure 1D, and similar for L133-138: refer

to Figure 1C.) Also rearrange figures so that those mentioned first appear earlier - i.e. Figure 1C and D referred to before figure 1B, Supp Fig 3B referred to before Supp Fig 3A.

- L181: what about the other 5 from cheese A and 14 from cheese C?
- L223-241: might be good if it was visualised somehow? Also refer to previous figures in the text.
- L223-241: would be nice to show in a figure the species and strain-level dynamics from short reads mapped back to the genomic catalog
- L289-290: last part should be in the discussion and not the results. L359-363, these should be moved to the discussion as they are not results
- does the adjusted q-value indicate significance?
- L349, 351, 419: days 3, 5, 7, 21 – standardise L807-808: 3 strains from a previous study, were these not found in this study? Is there a reason they were added in?
- Standardise - Dereplicate or de-replicate?
- In general, the font size in the Figures could be increased, quite hard to read at 100%
- Supplementary tables – lines are messy, can be cleaned up and fonts can be standardised
- Figure 5B: might be good to underline the genomes from this study
- Supp figure 12 can be made an actual figure, not in supplementary
- Supp table 19: please check day 0 replicate 1 for both bacteria and fungi

**Comments from the authors:**

We would like to thank the reviewers for their thoughtful comments and suggestions. We addressed them
as outlined below and believe they make the manuscript stronger overall.

**Reviewer #1 (Comments for the Author):**

The manuscript by Saak et al. deals with the investigation of the cheese rind microbiome and its genomic
potential and temporal dynamics. The work has been performed using the latest short- and long-
sequencing technologies and bioinformatics tools. The work is interesting, and the amount of data and
results generated is incredible. The different approaches (culture-independent, culture-dependent, in vitro
washed rind cheese system, pangenome) provide a good understanding of this complex microbiome from
different angles. The following comments/suggestions must be considered before the manuscript can be
accepted.

**Response:**

We thank the reviewer for this overall assessment. We agree that the different approaches employed in
this work contribute to a well-rounded understanding of the investigated microbiomes. We also thank the
reviewer for the helpful comments and suggestions. We have addressed them as detailed below.

**Comment 1.1:**

The discussion should be improved and extended to discuss more potential intrinsic and extrinsic factors
that could have driven the microbial ecology on the rind, as the authors suggest in the introduction. What
has caused this microbiome to shift over time? For example, there is no mention of changes in salt
concentration or pH during cheese ripening which, for sure, impact the rind microbiome. How was the
cheese salted? Dry or brine salting? The salt diffuses into the cheese over time. Was this considered? pH
changes during maturation are also important and need discussion.

**Response 1.1:**

Thank you very much for this suggestion. We agree that there are many intrinsic and extrinsic factors that
could drive the microbial ecology observed in this study. While the investigation of these factors was
beyond the scope of this work, we think that the developed *in vitro* community will provide an excellent
testing ground to delve into these questions. As such, we added the following sentences to the discussion
(Lines 587-594):

“This model community will allow the further investigation of potential intrinsic and extrinsic
factors that drive the microbial ecology within these type of communities. For example, surface moisture
and pH have been shown to impact community composition when comparing natural, bloomy and
washed-rind cheeses, while the salt concentration was found to be less of a predictor (5). Nevertheless,
given high exposure to salt during the ripening of washed-rind cheeses, this factor, too, should be
examined more closely in these communities. As the three communities examined in this study are
objected to brine salting, the interplay between salt content and surface moisture is another aspect that
should be investigated *in vitro*.”

**Comment 1.2:**

I am also wondering if the pangenome analysis of the *Psychrobater* needs to be included in this work. The
results are exciting and confirm what was already found by other results in the paper, but in my opinion,
these data should be explored more and discussed further. The authors described this part as "an
example," but I think these results have a more significant potential if exploited more. Can these data be
presented in more detail in another paper?

**Response 1.2:**

We agree with the reviewer that the pangenome analysis could be presented in more detail in another
paper. However, none of the authors are in a position to pursue this any further. Since we still want to

make this interesting result available to the wider scientific community we decided to keep it in this
manuscript, but streamlined the pangenome analysis.
As such, Fig. 5A and B now present a simplified version of the pangenome analysis of *Psychrobacter*
species and functional enrichment in cheese-associated *Psychrobacters* (also in light of comment 1.3).
Specifically, we removed the Anvi'o plot (original Fig. 5A) and moved the histogram (also original Fig.
5A) into the supplement (now Fig. S9). The phylogenetic tree is now Fig. 5A and the functional
enrichment analysis is now Fig. 5B. We removed the alignment of the type VI secretion clusters
altogether (originally Fig. 5D), including the corresponding text in the manuscript. While this finding is
highly interesting, we believe that it leads to many justified follow-up questions, which cannot be
satisfyingly answered within the scope of this work. Instead, we amended this figure with the pangenome
analysis of *Psychrobacter* isolates from this study (originally Fig. 6C and D) and use it as a lead-in into
the section describing the *in vitro* community reconstruction.

**Comment 1.3:**

As I said, much work has been done, and I understand the difficulties in presenting all the data in detail in
one manuscript. However, some figures are too complex, and the visualization is difficult to follow-for
example, figures 3A, 5A, and 5D.

**Response 1.3:**

We also agree with the reviewer that some of our figures are rather complex in an effort to convey all the
interesting findings. In response, we have removed Fig. 5A and D (see response 1.2). Regarding Fig. 3A,
while we understand that this is an information-rich figure, we consider it one of the keystone figures of
the paper that motivates many of the follow-up analyses. As such, we regard it as an important resource
for the reader that contains all relevant information in one location allowing the reader to reference this
figure while reading the manuscript.

**Minor comments**

**Comment 1.4:**

Several references are given at the same point. I think a maximum of 3 references are enough for each
statement. The introduction can be extended.

**Response 1.4:**

In an effort to be both as inclusive as possible in assigning credit to previous work done in the field, while
keeping the introduction streamlined, we indeed starkly summarized the literature in multiple locations
resulting in more than three citations given for several statements. We agree with the reviewer that this
deserves more detail and thus we expanded the introduction accordingly, thereby breaking up the cited
literature into more fine-grained statements about the state of the field.

**Comment 1.5:**

Line 88: Remove "genome". Plasmids are not associated with host genomes but with the host.

**Response 1.5:**

Thank you very much for this suggestion. We corrected this in the text.

**Comment 1.6:**

Line 575: remove "of washed-rind cheese communities". Not necessary here.

**Response 1.6:**

We agree with the reviewer and removed the unnecessary words.

**Comment 1.7:**
Line 618: Add the species name of the CCS196 if available

**Response 1.7:**
Thank you very much for this suggestion. We now added “(*Psychrobacter sp.*)” to the text as this is
currently the highest level of taxonomic classification available for this isolate.

**Comment 1.8:**
Line 370: is Fig 5A the correct number?

**Response 1.8:**
Thank you very much for catching this mistake on our part. It should refer to Fig. 6B based on the new
figure arrangement, not 5A. We corrected it in the text and also added a reference to the relevant
supplemental figure and table. The full sentence (lines 423-428) now reads:

“The removal of *G. geotrichum* favored the relative growth of the Actinobacteria over the
Proteobacteria (Fig. 6B, Fig. S10B, see Table S19 at <https://doi.org/10.5061/dryad.bg79cnpd8>), even
though overall bacterial absolute abundance was very similar between these two samples (Fig. S10A, see
Table S18 at <https://doi.org/10.5061/dryad.bg79cnpd8>)”.

**Comment 1.9:**
The author should give more information on the cheeses that were used for the study. For example,
mention production technologies used and other parameters which would help understand changes in the
rind microbiome as mentioned above (water content or dry matter of the fresh cheese, salt, ph, ripening
temperature, humidity, starter culture used ...).

**Response 1.9:**
Thank you very much for this suggestion. Table S1 (see at <https://doi.org/10.5061/dryad.bg79cnpd8>)
contains information about the wash and ripening schedules of Cheeses A, B and C. Unfortunately, we
cannot state more detailed information, e.g. regarding the starter cultures used, as this would infringe
upon the cheese makers proprietary information. However, as stated in response 1.1 we are aware that
environmental factors such as salt content, pH, ripening temperatures and humidity could affect the
communities’ successional dynamics. These factors can be tested using the *in vitro* community developed
in this work.

**Comment 1.10:**
Was there any possibility of cross-contamination between the cheeses during ripening since they were
ripened in the same facility?

**Response 1.10:**
Thank you, this is a great question. Indeed, there is a possibility of cross-contamination between the
cheeses as they are ripened in the same facility. This was one of the reasons behind the ANI analysis
presented in Fig. 3B, Fig. S4 and Table S10. To clarify this point, we have now added the following
sentences to the discussion (lines 517-525):

“When comparing the high-quality MAGs (and isolate genomes) between the three cheeses we
find that they contain both unique genomes as well as highly similar genomes (Fig. 3B, Fig. S4, see Table
S10 at <https://doi.org/10.5061/dryad.bg79cnpd8>). This finding is not surprising: on the one hand, we
expected the three cheeses to share community members as they are ripened in the same facility and
handled by the same cellar employees. The communities thus likely experience cross-inoculation during
ripening. On the other hand, the milk sources, milk processing techniques (pasteurized, non-pasteurized),
as well as wash schedules and ripening times (see Table S1 at <https://doi.org/10.5061/dryad.bg79cnpd8>)
differ between the cheeses. As such, it is unsurprising that they also each contain unique community
members “

**Comment 1.11:**
Line 470. Cheese rind was stored for up to 72h before DNA extraction in addition to the 48 h of cheese
storage. Could these two storage times impact the community composition and, therefore, functionality
described after metagenomic analysis?

**Response 1.11:**
Thank you very much for this question. These varying storage times resulted from complications due to
the COVID-19 pandemic (specifically shipment delays of the samples and reagents and restrictions on on-
site work). However, based on the results from the amplicon sequencing of three batches of each cheese,
we are confident that these variations in storage time did not significantly impact our results.

**Comment 1.12:**
DNA extraction was done only with lysozyme and proteinase K. This introduces a bias. Did the authors
check this? A combination of mechanical disruption and enzymatic treatment of the microbial pellet is
suggested in microbiome studies. Alternative two different DNA extraction methods are a better
alternative for the two different sequencing technologies (short and long DNA sequencing).

**Response 1.12:**
We agree with the reviewer that enzymatic lysis alone would not have been sufficient for these
communities. Indeed, we employed an initial mechanical disruption prior to incubation in the lysis buffer.
We now emphasize this in the text more clearly and changed the initial sentence in the methods section
(“For the DNA extraction, the rind samples were ground into powder in liquid nitrogen”) to “For the
DNA extraction, the rind samples were first disrupted mechanically by grinding into powder in liquid
nitrogen using mortar and pestles.” (lines 616-614)

**Reviewer #2 (Comments for the Author):**

This paper is a complex study that incorporates various methods to understand succession dynamics in
cheese rind microbiomes at various levels of resolution. It could be improved **with more discussion**, as it
is currently very result-heavy. Further discussion on the potential application of these methods, in
particular, the benefits and uses/application of a genomic catalog, the comparisons with existing studies
on the cheese rind microbiome and with other studies that have employed such methods would elevate the
discussion. There are too many supplementary figures and tables that can be removed (supp table 8 and
13) or combined (supp figure 1-2, supp tables 9-11).

**Response:**
We thank this reviewer for their thoughtful comments and suggestions. We have expanded the discussion
as outlined above and below. Specifically, we compared our results to those of existing studies (see, for
example, response 2.3) and we expanded the discussion with examples other microbial genomic catalogs,
the methods used to generate them and their potential applications (see, for example, response 2.8).

We also followed the advice of the reviewer and streamlined the supplemental tables by
combining Tables S9-S11. While the information in Table S8 can also be obtained through the respective
SRA submissions, we think it is convenient for the reader to be able to access that information in one
concise table that is part of this manuscript. As such, we decided to keep Table S8. We also decided to
keep Table S13, now Table S11. The range of the percentages of mapped reads is quite large, but not
uniformly distributed. The differences in mapping have a biological cause and actually provide valuable
inside into the robustness of the genomic catalog. This is directly relevant to comment 2.8. Regarding the
suggestion of combining Fig. S1 and S2, we argue that combining the two figures could lead to confusion
since Fig. S1 is an overview figure of the study while Fig. S2 illustrates the potential cause of batch-
irregularity of the fungal amplicon data presented in Fig. 1. However, we combined it with the original
Fig. S3 since both supplemental figures relate to the amplicon analysis. As such we combined Fig. S2 and

Fig. S3 into Fig. S2. We also combined additional supplemental figures and now have a total of 10
supplemental figures that accompany the manuscript.

Our responses to all other comments are outlined below.

**Comment 2.1:**

While the consistent successional patterns were evident, were there any differentially abundant
taxa/genera (bacteria/fungal) between batches? (L29-31)

**Response 2.1:**

Thank you very much for this question. Indeed, one batch of Cheese B showed the presence of the fungus
*Fusarium*, while the other two batches did not. In the text (starting in line 142) we describe with

“Principal component analysis based on Bray-Curtis indices indicates that while the different
cheeses are highly similar at the earliest sampled time points, they diverge from each other along
reproducible trajectories throughout aging (Fig. 1B). One exception is one batch of Cheese B, which
clusters more closely to Cheese A at the final fungal sequencing time point. This difference is mainly due
to *Fusarium*, which is found in all 3 batches of Cheese A and detected in the ITS sequences of that batch
of Cheese B, but not the other two batches of that same cheese (Fig. 1D). Indeed, a wheel from this batch
has a visibly different rind than cheese wheels from the other two batches (Fig. S2A).”

After personal communication with the cheese makers, we deduced that this difference is not
unusual for this cheese and depends on the affinage process. For example, the amount of time between the
last wash and the packaging affects the establishment of *Fusarium* in these rinds. As this information
pertains to the production processes developed by the cheese makers we do not go into this in detail in the
paper. Furthermore, since this occurrence does not change the overall conclusion of reproducible
succession patterns, we did not refer to it in the abstract.

**Comment 2.2:**

Was the long read data in Figure 2 and Supp figure 5 in agreement with the amplicon data in Figure 1?
Were there any differences?

**Response 2.2:**

This comment was addressed in an addition to the discussion (lines 449-486):

“Together, the amplicon sequencing (Fig. 1B-D) and long-read taxonomic classification (Fig. 2, Fig.
S3B) show that the mature communities of the investigated cheeses are dominated by bacteria, in
particular Proteobacteria, and they point to overall similar bacterial succession dynamics: both show that
Cheese A experiences relatively little taxonomic turnover between weeks 2 and 13 and that this
community is consistently dominated by *Psychrobacter*, *Pseudoalteromonas* and other Proteobacteria. In
contrast, Cheese B shows more of a successional turnover with *Psychrobacter* and *Pseudoalteromonas*
initially being of comparable abundance within the community until *Pseudoalteromonas* eventually drops
in abundance, while *Psychrobacter* and *Halomonas* become the abundant community members. For
Cheese C, the amplicon and long-read-based taxonomic classifications again reveal the same trend. Both
show a gradual takeover of the community by *Halomonas* with a concomitant rise of Gram-positive taxa
such as *Brevibacterium*. Notably, there were also differences regarding the results from the amplicon and
long-read-based classification. While many taxa were detected above the threshold by both techniques,
each also resulted in unique taxa. For example, *Glutamicibacter* was detected above the threshold using
the long-read sequencing, but not by amplicon sequencing. Similarly, while the Bacteroidetes
*Sphingobacterium* was detected by long-read sequencing, the Bacteroidetes *Mesonina* and *Myroides* were
detected by amplicon sequencing. Interestingly, all three genera are represented in our genome catalog
(Fig. 3A, see Table S9 at <https://doi.org/10.5061/dryad.bg79cnpd8>). These differences are likely caused
by biases inherent to the different sequencing techniques, such as uneven amplification of the 16S target
gene from different taxa, as well as differences in the databases used. While the amplicon data was
analyzed using the Greengenes database (29), the long read amplicon data was classified using the NCBI
nt database. That the genome catalog contains MAGs of those taxa detected by both classification
techniques as well as of those taxa detected by either of the techniques points to the fact that both

techniques capture part of the ground truth and that one technique alone is not sufficient to fully
characterize the taxonomic diversity of these techniques. Considering the fungi, the amplicon and long-
read data again contain similarities and differences, likely due to the same causes as for the bacterial part
of the communities. While the amplicon data, by nature of the technique, does not reveal details about the
relative abundances of fungi and bacteria in the communities, the long-read data shows that the
communities are already dominated by bacteria in week 2. The only exception is Cheese B, which is
dominated by *Debaryomyces* in week 2. Other Debaryomycetaceae, Saccharomycetales as well as
*Geotrichum* make up the majority of the rest of the community at this timepoint. In contrast, the amplicon
data suggests that the fungal communities of Cheese B at that timepoint are dominated by *Dipodascus*,
while *Debaryomyces* constitutes a smaller, yet substantial, part of the fungal communities in Cheese B in
261 week 2. Again, these differences are likely to be due to amplification biases in the amplicon sequencing as
well as database differences.”

**Comment 2.3:**

There were taxa that were specific to certain cheeses and some that were shared between all cheeses -
have these been seen in other studies? Elaborate more on this in the discussion, with examples.

**Response 2.3:**

Thank you very much for this question. The first part of this question, the degree to which cheeses contain
unique and shared taxa is specifically addressed in the ANI analysis of the MAGs in the respective
genomic catalogs, which is presented in Fig. 3B and the corresponding Fig S4 and Table S10. Please see
the following text from the manuscript (lines 211-225):

“Since the amplicon sequencing data as well as the long-read data point to an overlap between the
genera present in the three communities, we next tested whether identical or highly similar genomes were
recovered from these communities as well. Indeed, comparing the genomic catalogs recovered from the
three cheeses using ANI values reveals that the cheeses contain both common and unique genomes (Fig.
S4, see Table S10 at <https://doi.org/10.5061/dryad.bg79cnpd8>). Specifically, when considering an ANI
cut-off of 99, we observed that six MAGs were represented in the genomic catalogs from all three cheeses
(Fig. 3B). Based on GTDB-Tk (24), these MAGs were annotated as *Mesonnia* sp., *Vibrio casei*,
*Pseudoalteromonas nigrifaciens*, *Psychrobacter alimentarius*, *Vibrio litoralis*, and *Pseudoalteromonas*
*pyrdzensis*. Additionally, Cheeses A and B have 8 MAGs (and isolate genomes) in common, Cheeses A
and C share 2 MAGs and Cheeses B and C share 5 MAGs (and isolate genomes) (Fig. 3B, Fig. S4, see
Table S10 at <https://doi.org/10.5061/dryad.bg79cnpd8>). Lastly, Cheese A has 1 unique MAG, Cheese B
has 8 unique MAGs (and isolate genomes) and Cheese C has 24 unique MAGs (Fig. 3B, Fig. S4, see
Table S10 at <https://doi.org/10.5061/dryad.bg79cnpd8>).”

The second part of this question – whether the taxa detected in this study have been seen in other
studies - has now been elaborated on in the discussion (lines 487-496):

“Overall, our taxonomic classification results are similar to what was found for other washed-rind
cheeses. For example, the communities investigated here are dominated by bacteria, while fungi make up
a much smaller proportion of the community by the end of ripening (see (21, 22). In addition, we detect
very similar bacterial and fungal genera as previous studies investigating the composition of washed-rind
cheeses. For example, some taxa that were previously identified in washed-rind cheeses that were also
identified in this study include the Proteobacteria *Psychrobacter* and *Halomonas* and the Actinobacteria
*Brevibacterium* and *Glutamicibacter* (15, 17, 21, 22) as well as the fungi *Debaryomyces*, *Geotrichum* and
*Fusarium* (21–23). However, our multi-platform metagenomics approach provides new insights into
species and strain-level diversity and dynamics.”

**Comment 2.4:**
GTDB was used for MAGs - taxa differs in GTDB and NCBI databases - why wasn't NCBI used with
MEGAN, which was used for long read taxonomic classification?

**Response 2.4:**
GTDB-Tk was used for the classification of MAGs since the HiFi-MAG-Pipeline, which is part of the pb-
metagenomics-tools suite (<https://github.com/PacificBiosciences/pb-metagenomics-tools>), uses GTDB-
Tk. To better highlight that everything from binning to quality control and taxonomic classification was
done by the HiFi-MAG-Pipeline, we added “In the last step of the pipeline” to the initial last sentence of
the corresponding methods section (Lines: 738-741: “In the last step of the pipeline, the high-quality
MAGs are then analyzed using the Genome Taxonomy Database Toolkit (GTDB-Tk)(24), which attempts
to identify the closest reference genome and assign taxonomy for each MAG.”). To be consistent, we then
also used GTDB-Tk to classify the taxonomies of the final genomes in the genomic catalog. Conversely,
the Taxonomic-Profiling-Nucleotide pipeline from the same pb-metagenomics-tools suite utilizes the
NCBI nt database. To ensure reproducibility of our results, we chose not to change individual components
of the two utilized tools (HiFi-MAG-Pipeline and Taxonomic-Profiling-Nucleotide pipeline).

**Comment 2.5:**
While Figure 3 is a very nice figure, it might be clearer if expressed in a table, like a summary table of the
MAGs selected for the genomic catalog.

**Response 2.5:**
We agree that summary tables of the high-quality MAGs in the genomic catalog are helpful in
interpreting Fig. 3. We kindly point to Table S9 (see at <https://doi.org/10.5061/dryad.bg79cnpd8>), which
provides a summary of the information in Fig. 3A including the MAG designation, bin name, circularity,
number of contigs, size, completeness and contamination estimations as well as taxonomic classification.
The results of Fig. 3B can be found in Table S10.

**Comment 2.6:**
Not so clear what the purpose of Supp table 12 - if it is to just mention that several bins contain
similar/identical genomes? Or that isolates and assembled MAGs are the same?

**Response 2.6:**
Thank you very for pointing this out. Table S12, now Table S10, shows the comparison of the three
genomic catalogs based on ANI values. As detailed in response 1.10, we wanted to test to which degree
the same or highly similar genomes were recovered from the three cheeses. As we carried out the ANI
analysis with the genomes from the genomic catalogs, the redundancy within each catalog has already
been removed and was not subject of this analysis. To better clarify the purpose of this analysis, we
adapted the text as follows:

Initially: “Consistent with our amplicon sequencing data, comparing the genomic catalogs
recovered from the three cheeses using ANI values reveals that the cheeses contain both common and
unique genomes (Suppl. Fig. 12).”

Now (lines 211-216): “Since the amplicon sequencing data as well as the long-read data point to
an overlap between the genera present in the three communities, we next tested whether identical or
highly similar genomes were recovered from these communities as well. Indeed, comparing the genomic
catalogs recovered from the three cheeses using ANI values reveals that the cheeses contain both common
and unique genomes (Fig. S4, see Table S10 at <https://doi.org/10.5061/dryad.bg79cnpd8>).”

The results of this ANI analysis are then summarized in Fig. 3B.

**Comment 2.7:**
Supp table 13 - can just give a range in the text, don't need another table.

**Response 2.7:**
Thank you very much for this suggestion. However, as is described, e.g. in response 2.8, this data is
important in evaluating the completeness of the genomic catalog and we thus opted to keep Table S13,

now Table S11.

**Comment 2.8:**

Genomic catalog - How should the robustness of the genomic catalog be evaluated? - are there other
papers that specify this? (L220-222). It would also be nice to give examples of the use of this in future
studies or cite papers that have done so.

**Response 2.8:**

To address this comment we expanded the discussion with examples of studies employing similar
approaches compared to ours for the generation of genomic catalogs and using metaHi-C to detect
putative HGT events (lines 526-538):

“Our genomic catalog joins the ranks of genomic catalogs previously created, such as the recently
created catalog of reference genomes from the human gut microbiome (31), which combined MAGs and
isolate genomes, the Skin Microbial Genome Collection, which also integrates isolate cultivation with
metagenomics (32) as well as other dairy microbial gene catalogs, such as one of 117 isolate genomes
(30). These genomic catalogs can be used to develop testable hypotheses, design genetic tools and high-
throughput screening experiments that require reference genomes and provide reference genomes for
additional omics approaches, such as proteomics. In this study, we utilized the bacterial genomic catalog
to further investigate the biology within these microbiomes including the potential for horizontal gene
transfer and the enrichment of specific functions. First, we leveraged the long reads together with metaHi-
C reads to associate putative mobile genetic elements with their hosts. A similar approach was used in
(33), where Hi-C was used to associate MGEs with their bacterial hosts in the human gut microbiome and
track putative HGT events in longitudinal samples.”

We further provided more details in the discussion on the evaluation of the catalog’s completeness
and robustness (lines 504-516):

“To assess the completeness of our genomic catalog we mapped short reads from the same
communities back to the catalog (see Table S11 at <https://doi.org/10.5061/dryad.bg79cnpd8>). The
percentage of reads mapped to the catalog at each timepoint gave an indication for how well the catalog
represents the community at the respective timepoints. Indeed, several timepoints for Cheeses B and C
achieved read mapping percentages over 90% suggesting that the genomic catalog captures the
community diversity well. In contrast, only 19% of the reads from week 2 of Cheese B mapped to the
respective genomic catalog. However, this made sense given that at this timepoint this community is
dominated by fungi, which were not included in our catalog. In future studies, the robustness of these
genomic catalogs can be estimated by mapping reads from novel washed-rind cheese communities and
determining to which degree our generated catalog is representative of these types of communities in
general. A similar approach was followed for a previously generated genomic catalog of dairy microbes
(30).”

**Comment 2.9:**

Why was *Psychrobacter* chosen for pangenome analysis and what is the importance of the co-existence of
multiple *Psychrobacter* spp.?

Also the categories of the other genomes included (supp table 17) were a little strange/inconsistent? (e.g.
Marine sediment and lake water classified as misc. and not marine? Lake water as misc)

**Response 2.9:**

In order to clarify why *Psychrobacter* was chosen for the pangenome analysis we expanded the
discussion, which now reads (lines 553-563):

“Next, we investigated the striking diversity of *Psychrobacter* within the communities. Genus-
level diversity has been observed in other microbial communities, including cheese rinds, and their study
*in vitro* has proved valuable. For example, by studying closely related, frequently co-occurring species of
*Staphylococcus* the importance of biotic interactions with fungi was revealed (7). Since in the
communities investigated here *Psychrobacter* showed the greatest sub-genus diversity both in the
genomic catalog as well as among the isolates from Cheese B, we first followed up on this diversity

through leveraging the genomic catalog. Specifically, we investigated functional enrichment within
cheese-associated *Psychrobacter* MAGs and isolate genomes as compared to *Psychrobacter* genomes
associated with other environments (Fig. 5B). This question was of particular interest as *Psychrobacters*
are found in a variety of environments, both free-living and host-associated, and that they show habitat-
dependent adaptations (34)

Regarding the choice of categories, we thank the reviewer for pointing out the inconsistencies.
We have now corrected this and reclassified the isolates from tidal flat sediment, deep sea sediment and
marine sediment as “marine”. We adjusted the color scheme in the phylogenetic tree in Fig. 5A (formerly
Fig. 5B) accordingly and also fixed Table S15 (formerly Table S17). The functional enrichment analysis
was not affected by this reclassification as the enrichment analysis was done comparing cheese vs non-
cheese.

**Comment 2.10:**

L366-382 and Supp table 19: these results might be better visualised as a figure, instead of Supp table 19.
If they are already in figures, please refer to them. It also might be good to combine a few of the relevant
ones into one supp figure.

**Response 2.10:**

Thank you very much for bringing this to our attention. We made a typing error here. In this section we
actually refer to Fig. 6B, Fig. S10B and Table S19 (according to the new figure and table arrangement)
instead of Fig. 5A and Table S19. Figures 6B and S10B are the visualizations of the data contained in
Table S19. We fixed these errors. The text now reads (lines 423-430):

“The removal of *G. geotrichum* favored the relative growth of the Actinobacteria over the
Proteobacteria (Fig. 6B, Fig. S10B, see Table S19 at <https://doi.org/10.5061/dryad.bg79cnpd8>), even
though overall bacterial absolute abundance was very similar between these two samples (Fig. S10A, see
Table S18 at <https://doi.org/10.5061/dryad.bg79cnpd8>). Absolute abundance based on read counts reveals
that the decrease in gamma-proteobacterial abundance is largely due to the poor growth of
*Pseudoalteromonas* (see Table S19 at <https://doi.org/10.5061/dryad.bg79cnpd8>).”

**Other more minor/specific comments:**

**Comment 2.11:**

As there are many figures and supplementary figures and tables, it would be good if they were referred to
in the text (e.g. L129-133: can add reference to Figure 1D, and similar for L133-138: refer to Figure 1C.)
Also rearrange figures so that those mentioned first appear earlier - i.e. Figure 1C and D referred to before
figure 1B, Supp Fig 3B referred to before Supp Fig 3A.

**Response 2.11:**

We edited the text to include references to the corresponding (supplementary) figures and tables wherever
relevant. We agree that this improves the clarity of the text. We also thank the reviewer for pointing out
the oversight on our part regarding the ordering of the figures and tables throughout the text. We now
fixed this issue throughout.

**Comment 2.12:**

L181: what about the other 5 from cheese A and 14 from cheese C?

**Response 2.12:**

We expanded on this section to give more detail on the rest of the bins for each of the three cheeses. The
text now reads (lines 203-210):

“For Cheeses A and C, we recovered 17 and 37 high-quality MAGs, respectively (Fig. 3, see
Table S9 at <https://doi.org/10.5061/dryad.bg79cnpd8>). In Cheese A, 12 out of the 17 high-quality MAGs
were both single-contig and circular while the other 5, non-circular bins contained 2-6 contigs each. In the
case of Cheese C, 24 of the 37 high-quality MAGs were single-contig and 19 of those were circular. The
rest of the bins from Cheese C contained 2-10 contigs each. For Cheese B, we recovered 11 high-quality

MAGs and 16 isolate genomes (see Table S9 at <https://doi.org/10.5061/dryad.bg79cnpd8>). Four out of the
11 high-quality MAGs were both single-contig and circular. The other 7, non-circular bins, contained 2-8
contigs each.”

**Comment 2.13:**

L223-241: might be good if it was visualised somehow? Also refer to previous figures in the text.

**Response 2.13:**

This is indeed visualized as part of Fig. 3A. We now specifically refer to this figure in the text. We have
further added a sentence to the figure legend to more clearly indicate that the mapping is visualized in the
Anvi'o figure and that the height of the bars indicates the number of reads mapped to that particular
genomic region:

Before: “To estimate relative abundances of MAGs over time, the short-reads from weeks 2, 3, 4,
9 and 13 were mapped to the genomic catalog for each cheese.”

Now: “To estimate relative abundances of MAGs over time, the short-reads from weeks 2, 3, 4, 9
and 13 were mapped to the genomic catalog for each cheese. The height of the bars indicates the number
of mapped reads to the respective genomic region.”

**Comment 2.14:**

L223-241: would be nice to show in a figure the species and strain-level dynamics from short reads
mapped back to the genomic catalog

**Response 2.14:**

Please refer to response 2.13.

**Comment 2.15:**

L289-290: last part should be in the discussion and not the results. L359-363, these should be moved to
the discussion as they are not results

**Response 2.15:**

Thank you for the suggestion. We moved the suggested part from L289-290 into the discussion where we
added the following section (lines 547-550):

“What is more, some of the elements identified in this study that contain iron-uptake pathways
also contain genes related to phosphanate transport. This association between horizontally-transferred iron
uptake regions and phosphanate transport genes has also been previously observed in cheese (28).”

However, we kept the statement from L359-363 in the results section as this statement is not a
discussion of our findings, but rather an explanation for the rationale behind the *in vitro* experiment set-
up.

**Comment 2.16:**

does the adjusted q-value indicate significance?

**Response 2.16:**

That is correct. A q-value is a p-value that has been corrected to control for false discovery rate. This is
important when performing multiple comparisons like we are in this analysis, where an enrichment test is
performed for each function separately. This correction is standard practice in statistics to avoid excessive
false positives when testing multiple hypotheses simultaneously and the reason why q-values, rather than
p-values, are reported for this analysis in Table S16.

**Comment 2.17:**

L349, 351, 419: days 3, 5, 7, 21 - standardise

**Response 2.17:**

Thank you very much for pointing this out. This has been fixed in the text.

**Comment 2.18**
L807-808: 3 strains from a previous study, were these not found in this study? Is there a reason they were
added in?

**Response 2.18:**

Thank you very much for this question. These strains were added in from previous studies to both ensure
that the community represents the breadth and depth of diversity of typical washed-rind cheese
microbiomes and to enable transferability of our results to other ongoing projects in our group.

Specifically, *Vibrio casei* JB196 and *Hafnia alvei* JB232 were added since both of these species were
represented in our genomic catalog, but not isolated during our isolation efforts. As such, we chose two
strains from our strain collection, which are regularly used in community experiments in our lab.

*Brevibacterium linens* JB5 was chosen in the same vain, as several representative of the genus
*Brevibacterium* were represented in the genomic catalog and JB5 is regularly utilized in our other
projects. To clarify these points we added the following sentences to this paragraph (lines 393-397):

“Isolates *Vibrio casei* JB196 and *Hafnia alvei* JB232 were chosen from our existing strain
collection, as these two species were represented in our genome catalog, but not isolated from the
communities as part of our isolation efforts. These two strains, together with the chosen *Brevibacterium*
*linens* JB5, were isolated as part of previous studies (5, 6) and are regularly utilized in our community
experiments.”

**Comment 2.19:**

Standardise - Dereplicate or de-replicate?

**Response 2.19:**

Thank you very much for catching this. This is been fixed to “dereplicate” throughout the text.

**Comment 2.20:**

In general, the font size in the Figures could be increased, quite hard to read at 100%

**Response 2.20:**

We agree with the reviewer and increased the font sizes in the figures wherever possible.

**Comment 2.21:**

Supplementary tables - lines are messy, can be cleaned up and fonts can be standardised

**Response 2.21:**

Thank you very much for bringing this to our attention. We have now standardized the fonts in all tables
to Helvetica, size 12. In addition, we have edited the tables to eliminate the need for lines and we have
streamlined Table S14 to be more easily interpreted.

**Comment 2.22:**

Figure 5B: might be good to underline the genomes from this study

**Response 2.22:**

Thank you for this suggestion. We agree it would be helpful to better indicate the genomes from this
study. To this end we added stars next to the respective genomes in Fig. 5A (formerly Fig. 5B).

**Comment 2.23:**

Supp figure 12 can be made an actual figure, not in supplementary

**Response 2.23:**

Thank you very much for this suggestion. To avoid redundancy with the figure legend in Fig. 6B, we
added a modified version of the former Fig. S12 to Fig. 6 (Fig. 6A).

**Comment 2.24:**
Supp table 19: please check day 0 replicate 1 for both bacteria and fungi
**Response 2.24:**
Thank you very much for catching this. Neither the bacterial nor fungal counts for Day 0 are available
and the table entries for both should be N/A. This has now been corrected.

November 20, 2022

Dr. Rachel J Dutton
University of California, San Diego
AP&M Building, room 3872
9500 Gilman Drive
La Jolla, CA 92093-0116

Re: mSystems00701-22R1 (Longitudinal, multi-platform metagenomics yields a high-quality genomic catalog and guides an *in vitro* model for cheese communities)

Dear Dr. Dutton (Rachel):

Your manuscript has been accepted, and I am forwarding it to the ASM Journals Department for publication. For your reference, ASM Journals' address is given below. Before it can be scheduled for publication, your manuscript will be checked by the mSystems production staff to make sure that all elements meet the technical requirements for publication. They will contact you if anything needs to be revised before copyediting and production can begin. Otherwise, you will be notified when your proofs are ready to be viewed.

Publication Fees:

If you would like to submit a potential Featured Image, please email a file and a short legend to mSystems@asmusa.org. Please note that we can only consider images that (i) the authors created or own and (ii) have not been previously published. By submitting, you agree that the image can be used under the same terms as the published article. File requirements: square dimensions (4" x 4"), 300 dpi resolution, RGB colorspace, TIF file format.

We recognize that the video files can become quite large, and so to avoid quality loss ASM suggests sending the video file via <https://www.wetransfer.com/>. When you have a final version of the video and the still ready to share, please send it to mSystems staff at mSystems@asmusa.org.

Sincerely,

Paul Cotter
Editor, mSystems

Journals Department
Fig.S8: Accept
Fig.S2: Accept
Fig.S6: Accept
Fig.S10: Accept
Fig.S3: Accept
Fig.S5: Accept
Fig.S9: Accept
Fig.S7: Accept
Fig.S1: Accept
Fig.S4: Accept